# *KVSink*: Understanding and Enhancing the Preservation of Attention Sinks in KV Cache Quantization for LLMs

**Zunhai Su**
Shenzhen International Graduate School, Tsinghua University
zh-su23@mails.tsinghua.edu.cn

**Kehong Yuan**[*]
Shenzhen International Graduate School, Tsinghua University
yuankh@sz.tsinghua.edu.cn

## Abstract

Key-Value (KV) cache quantization has become a widely adopted optimization technique for efficient large language models (LLMs) inference by reducing KV cache memory usage and mitigating memory-bound constraints. Recent studies have emphasized the importance of preserving the original precision of KVs for the first few tokens to ensure the protection of attention sinks. While this approach has proven effective in mitigating performance degradation, its underlying principles remain insufficiently understood. Moreover, it fails to address the recent discovery that attention sinks can emerge beyond the initial token positions. In this work, we elucidate the underlying mechanisms of attention sinks during inference by examining their role in the cross-layer evolution of extreme activation outliers. Additionally, we provide a comprehensive analysis of the interplay between attention sinks and KV cache quantization. Based on our enhanced understanding, we introduce *KVSink*, a plug-and-play method that effectively predicts sink tokens with negligible overhead, enabling more thorough preservation. Extensive experiments demonstrate that KVSink outperforms the existing Preserve-First-N (PFN) strategy, offering more effective preservation of attention sinks during KV cache quantization. Moreover, when applied to the well-established KVQuant method, KVSink further improves perplexity (PPL) and reduces reliance on 16-bit numerical outliers.

## 1 Introduction

Transformer-based (Vaswani et al., 2017) large language models (LLMs), including GPT (Achiam et al., 2023), LLaMA (Dubey et al., 2024), and DeepSeek (Liu et al., 2024a; Guo et al., 2025), have revolutionized various domains of artificial intelligence research, including natural language processing (Hadi et al., 2023; Zhao et al., 2023), computer vision (Zhang et al., 2024), and multimodal understanding (Liang et al., 2024). However, the impressive capabilities of LLMs come with significant challenges due to their extensive size and computational demands (Zhu et al., 2024), along with the substantial Key-Value (KV) cache generated during inference (Li et al., 2024a), all of which hinder their deployment and practical application. KV cache facilitates LLMs inference by avoiding recomputation of past KVs. However, as the batch size and context length increase, the oversized KV caches become a significant memory bottleneck (Liu et al., 2024c). KV cache compression has emerged as a promising direction to mitigate this challenge (Shi et al., 2024; Li et al., 2024a), encompassing a broad array of techniques—including quantization (Hooper et al., 2025; Su et al., 2025a), pruning (Xiao et al., 2023; Zhang et al., 2023), fusion (Liu et al., 2025;

---

[*]Corresponding author: Kehong Yuan

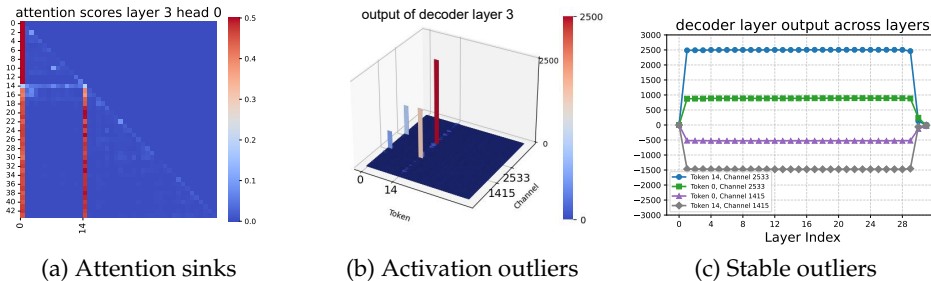

(a) Attention sinks  (b) Activation outliers  (c) Stable outliers

Figure 1: Visualizations of attention sinks and extreme activation outliers in LLaMA2-7B. (1a) illustrates the presence of attention sinks at tokens 0 and 14. (1b) shows extreme activation outliers in the output of the decoder layer 3, which emerge at attention sink tokens. (1c) shows the distribution of stable outliers across decoder layers. Unless otherwise specified, all visualizations use the following input from MMLU (Hendrycks et al., 2021): "The following are multiple-choice questions (with answers) about machine learning. \n\n..."

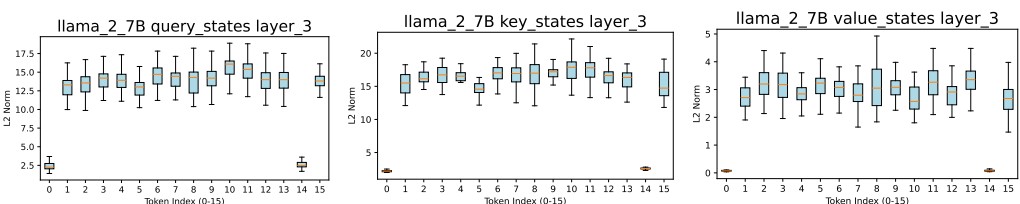

Figure 2: L2 norm distributions of Queries, Keys, and Values for all heads in the attention layer 3 of LLaMA2-7B across first 15 tokens. The sink tokens (0 and 14) display significantly smaller norms compared to non-sink tokens.

Wan et al., 2024), budget-aware allocation (Xiao et al., 2024; Cai et al., 2024), and low-rank decomposition (Chang et al., 2024; Saxena et al., 2024).

Xiao et al. (2023) characterizes the widely observed attention sink phenomenon, where LLMs tend to assign disproportionately high attention to the initial token, as shown in Figure 1a. Further research (An et al., 2025; Guo et al., 2024) suggests that attention sinks have a profound impact on the compression of LLMs. Specifically, the corresponding KVs require higher precision during quantization for adequate protection (Hooper et al., 2025; Su et al., 2025a; Duanmu et al., 2024; Su et al., 2025b) and must also be preserved during pruning (Xiao et al., 2023; Zhang et al., 2023). Additionally, as shown in Figure 1b, attention sink tokens have been found to exhibit extreme activation outliers (Bondarenko et al., 2021; Sun et al., 2024; An et al., 2025), which significantly impact activation quantization (Liu et al., 2024b; Li et al., 2024b). Although the practice of retaining the original precision of KVs for sinks tokens has proven effective in mitigating performance degradation, existing practices and understanding still have several limitations: *(1)* inadequate systematic analysis of attention sinks and their related phenomena during LLM inference; *(2)* lack of in-depth understanding of the mutual impact between attention sinks and KV cache quantization; *(3)* current implementations focus on statically persevering the KVs of first few tokens, which has proven insufficient in light of recent findings (Sun et al., 2024; Yu et al., 2024) indicating that attention sinks can occur at other positions, as shown in Figure 1a.

In this work, we aim to deepen the understanding of the role of attention sinks during LLM inference and their interplay with KV quantization, addressing the aforementioned limitations and advancing research in LLM compression and interpretability. We begin by elucidating the role of attention sinks through an examination of the cross-layer evolution of various types of extreme activation outliers in Section 3. Previous studies (Sun et al., 2024; Guo et al., 2024) have identified that extreme activation outliers manifest in the hidden states between decoders. As shown in Figure 1c, these outliers emerge and stabilize in the intermediate decoder layers, maintaining a persistent presence at sink tokens and LLM-specific channels, exhibiting large and consistent magnitudes. Given their distinctive properties and their central role in our analysis, we refer to them as ***stable outliers*** for clarity. Interestingly, we further observe that stable outliers follow a structured progression across decoder layers, evolving through the stages of emergence, stabilization, and dissipation,

driven by extreme activation outliers originating from the down-projection layer in the feed-forward network. Additionally, attention sinks emerge and persist throughout the stabilization stage, serving as the core mechanism for maintaining stability. As shown in Figure 2, this mechanism simultaneously imposes distinct numerical characteristics on the Queries, Keys, and Values of sink tokens, inevitably affecting KV quantization.

Then, in Section 4, we provide a comprehensive analysis of the interplay between attention sinks and KV cache quantization. We quantitatively assess the impact of attention sinks on various quantization schemes through quantization error analysis. Our study also confirms that quantization significantly disrupts the implicit attention biases introduced by attention sinks. Finally, based on our enhanced understanding, we introduce **KVSink**, a plug-and-play method that effectively predict sink tokens by leveraging the intrinsic relationship between attention sinks and stable outliers. Our contributions are summarized as follows:

• We advance the understanding of extreme activation outliers and attention sinks, elucidating the role of attention sinks during the stabilization phase of stable outliers, and clarifying the mechanism by which attention sinks influence KV cache quantization.

• To the best of our knowledge, this is the first work to thoroughly analyze and reveal the mutual influence between attention sinks and KV cache quantization. Our work not only deepens the understanding of attention sink preservation in KV cache quantization but also provides valuable insights for the development of more refined approaches in the future.

• Extensive experiments demonstrate that KVSink addresses the limitations of existing Preserve-First-N (PFN) strategy, providing more effective preservation with negligible overhead. Additionally, KVSink further refines the well-established KVQuant method, leading to improved perplexity (PPL) and reduced dependence on 16-bit numerical outliers.

## 2 Preliminary

**Transformer decoder.** LLMs are typically structured as a stack of Transformer decoder blocks (Vaswani et al., 2017), each consisting of a multi-head self-attention (MHSA) layer and a feed-forward network (FFN) layer. We use the widely adopted LLaMA (Dubey et al., 2024) architecture as an illustrative example, with a concise depiction provided in Figure 3. After tokenization and embedding, the input to the first decoder can be represented as $H^0 = \{h_1^0, h_2^0, \ldots, h_n^0\} \in \mathbb{R}^{n \times d}$, where $h_i$ denotes the $i$-th input token, $d$ is the embedding dimension, and $n$ is the length of the tokenized input sequence. Then, the output of the $l$-th decoder block, $H^l \in \mathbb{R}^{n \times d}$, is given by:

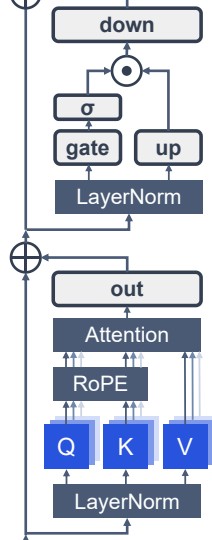

$$H^l = \text{FFN}\left(\text{LN}_{ffn}\left(H^{l'}\right)\right) + H^{l'}, \tag{1}$$

$$H^{l'} = O^l + H^{l-1}, O^l = \text{MHSA}\left(\text{LN}_{mhsa}\left(H^{l-1}\right)\right), \tag{2}$$

where $1 \leq l \leq L$, with $L$ denoting the total number of blocks. LN refers to layer normalization (with LLaMA employing pre-norm), $O^l$ representing the output of the MHSA, and $H^{l'}$ denoting the output of residual summations after the MHSA. Previous studies have found that Transformer-based models tend to learn structured activation outliers in $H^l$ and $H^{l'}$, which give rise to specific attention patterns that focus on special tokens, including BERT (Kovaleva et al., 2019), Vision Transformer (ViT) (Bondarenko et al., 2023), and LLM (Sun et al., 2024). A more detailed discussion of related works on this topic is provided in Appendix A.

Figure 3: Transformer decoder.

**Multi-head self-attention.** After LN, the input $H^{l-1}$ is projected through the weight matrices $W_Q^l, W_K^l, W_V^l \in \mathbb{R}^{d \times d}$ to generate the Queries, Keys, and Values, which are then divided into

$K$ heads, denoted as $Q^{l,k}, K^{l,k}, V^{l,k}$, for $1 \leq k \leq K$. The MHSA is computed as:

$$A^{l,k} = \text{Softmax}\left(\frac{Q^{l,k}K^{l,k^T}}{\sqrt{d_k}} + M\right), \quad O^l = \text{Concat}_{k=1}^K\left(A^{l,k}V^{l,k}\right)W_O^l, \tag{3}$$

where $M$ represents the attention mask, and $d_k = d/K$. For simplicity, the rotation position encoding (RoPE) (Su et al., 2024) applied to the Queries and Keys is omitted here.

**Feed-forward network.** Next, $H^{l'}$ is passed through a new LN and subsequently enters the feed-forward network layer:

$$H^l = \text{FFN}\left(\text{LN}_{ffn}(H^{l'})\right) + H^{l'} = \left(\sigma\left(\text{LN}_{ffn}(H^{l'})W_g\right) \odot \text{LN}_{ffn}(H^{l'})W_u\right)W_d + H^{l'}, \tag{4}$$

where $W_g$, $W_u$, and $W_d$ are the weight matrices for the gating, up-projection, and down-projection. $\sigma$ denotes the activation function, and $\odot$ represents the Hadamard product.

**KV cache.** The inference process comprises two stages: the prefill phase and the decoding phase. In the prefill phase, the LLM processes the token sequence generated from the input and produces the initial output token. Each attention layer $l$ computes and caches the KV tensors $K_{\text{cache}}^l$ and $V_{\text{cache}}^l$. In the decoding phase, the model takes the newly generated token as input. Let $t^l \in \mathbb{R}^{1 \times d}$ denote the input embedding of the $l$-th attention layer. Each attention layer computes the Queries, Keys, and Values $t_Q^l$, $t_K^l$, and $t_V^l$ as:

$$t_Q^l = t^l \cdot W_Q^l, \quad t_K^l = t^l \cdot W_K^l, \quad t_V^l = t^l \cdot W_V^l. \tag{5}$$

Then, $t_K^l$ and $t_V^l$ are used to update the KV cache, with the complete KV supporting subsequent MHSA computations:

$$K_{\text{cache}}^l \leftarrow \text{concat}(K_{\text{cache}}^l, t_K^l), V_{\text{cache}}^l \leftarrow \text{concat}(V_{\text{cache}}^l, t_V^l). \tag{6}$$

The growing size of the KV cache presents significant challenges in terms of memory usage and access latency, underscoring the need for efficient compression. Low-bit quantization has emerged as an effective approach, with related works discussed in Appendix B.

## 3 Attention Sinks and Extreme Activation Outliers

To enhance the understanding and preservation of attention sinks, we first present our findings on the intrinsic relationship between the cross-layer evolution of different types of extreme activation outliers in Section 3.1. Subsequently, we elucidate the core mechanism through which attention sinks maintain the stability of stable outliers in Section 3.2.

### 3.1 Cross-Layer Evolution of Extreme Activation Outliers

We begin by specifying the activations in which each type of outlier occurs, using the same notation as in Section 2:

(1) The input to the down-projection layer, denoted as $X_{\text{d\_in}}^l$, is given by

$$X_{\text{d\_in}}^l = \sigma\left(\text{LN}_{ffn}(H^{l'})W_g\right) \odot \text{LN}_{ffn}(H^{l'})W_u, \tag{7}$$

(2) The output of the down-projection layer, denoted as $X_{\text{d\_out}}^l$, is given by

$$X_{\text{d\_out}}^l = X_{\text{d\_in}}^l W_d, \tag{8}$$

(3) The output of the residual summation after the MHSA, denoted as $H^{l'}$,

(4) The output of the residual summation after the FFN, denoted as $H^l$, where **stable outliers** emerge. Notably, it satisfies $H^l = X_{\text{d\_out}}^l + H^{l'}$.

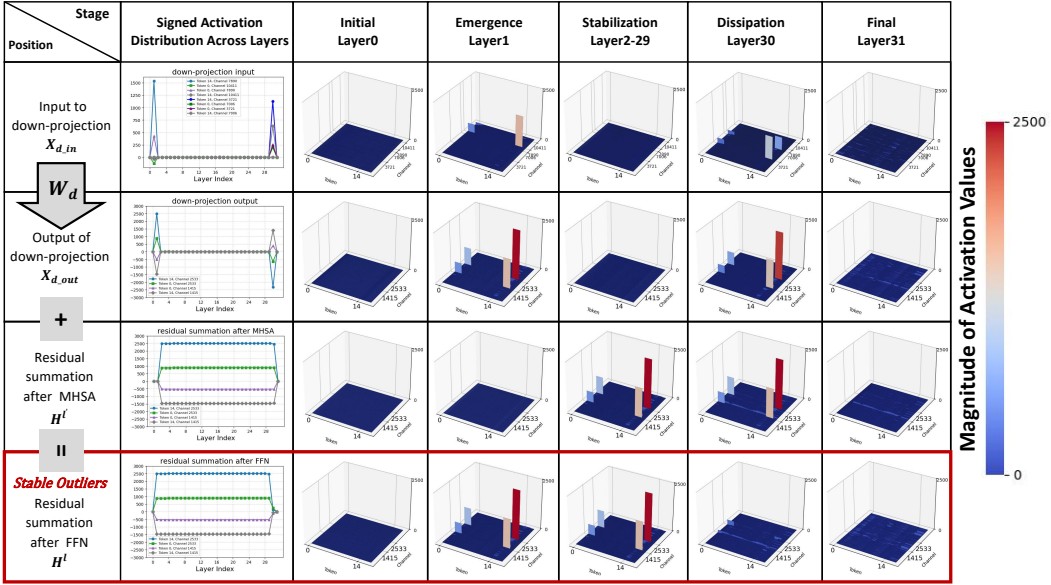

Figure 4: Visualizations of various types of extreme activation outliers in LLaMA2-7B. The left side illustrates the cross-layer distribution of various types of extreme activation outliers, while the right side demonstrates their behavior in different stages.

While all these outliers appear at attention sink tokens and exhibit significantly larger magnitudes, they differ in their locations within the model and follow distinct cross-decoder-layer distribution patterns. As shown on the left side of Figure 4, outliers in $X_{\text{d\_in}}^l$ and $X_{\text{d\_out}}^l$ appear in the early and late layers, while outliers in $H^{l'}$ and $H^l$ remain present across the intermediate layers. Interestingly, we found that the cross-layer evolution of these outliers unveils intrinsic relationships. Stable outliers, driven by outliers in $X_{\text{d\_in}}^l$ and $X_{\text{d\_out}}^l$, undergo a structured progression, which we categorize into five stages: initial, emergence, stabilization, dissipation, and final (as shown on the right side of Figure 4).

In the initial stage, no noticeable outliers are present. Then, in the emergence stage, extreme outliers first appear in $X_{\text{d\_in}}^l$, subsequently propagating to $X_{\text{d\_out}}^l$, and ultimately inducing the emergence of stable outliers in $H^l$ through the residual connection. The stabilization stage extends across the intermediate layers, encompassing the majority of the decoder layers. In this stage, outliers persist in $H^{l'}$ and $H^l$, while $X_{\text{d\_in}}^l$ and $X_{\text{d\_out}}^l$ no longer exhibit extreme outliers. Instead, their magnitudes decrease significantly, resulting in minimal contribution to the variation in extreme outliers. During the dissipation stage, extreme outliers re-emerge in $X_{\text{d\_in}}^l$, and their propagation to $X_{\text{d\_out}}^l$ generates outliers at the same positions as those observed during the emergence stage, with similar magnitudes but opposite signs. After the residual summation, this results in a significant reduction or disappearance of the stable outliers. Finally, in the last stage, no noticeable extreme outliers remain, and the model is ready to generate the output. Overall, these various extreme activation outliers demonstrate systematic relationships and interactions. Experimental results on additional models and inputs can be found in Appendix D.

## 3.2 Attention Sinks and the Stabilization of Stable Outliers

A notable characteristic of stable outliers is that during the stabilization phase, they remain consistently present, with their values varying only slightly. This implies that during the stabilization phase, the FFN and MHSA layers make minimal updates to the hidden states. Motivated by this and drawing inspiration from previous research (Bondarenko et al., 2023), we conclude that LLMs leverage attention sinks to achieve this behavior in the MHSA layer. Specifically, this process is governed by the following two key mechanisms:

**QKV suppression.** As shown in Figures 2 and 5, the Queries, Keys, and Values of sink tokens exhibit significantly smaller norms compared to non-sink tokens.

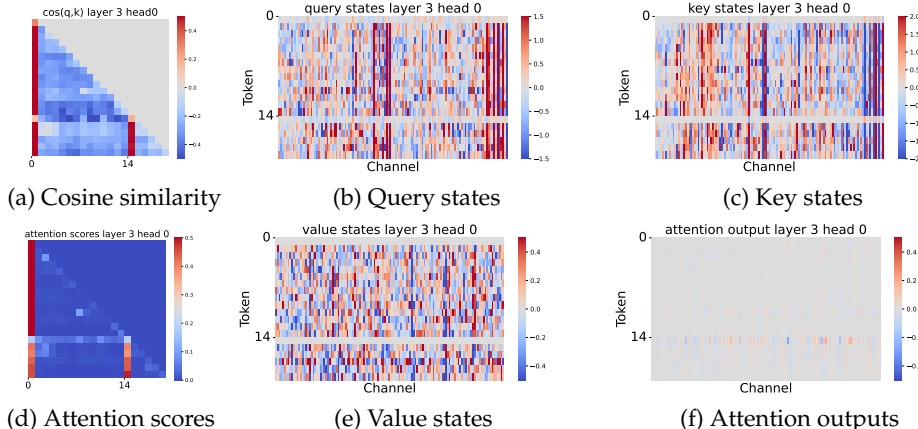

(a) Cosine similarity     (b) Query states     (c) Key states

(d) Attention scores     (e) Value states     (f) Attention outputs

Figure 5: (5b), (5c), and (5e) illustrate QKV suppression. (5a) highlights the high cosine similarity of QK. (5f) visualizes the attention output.

| Quantization Schemes | | Per-Token Key Quantization Error | | | Per-Token Value Quantization Error | | | Per-Channel Key Quantization Error | | |
|---|---|---|---|---|---|---|---|---|---|---|
| | Bits | overall | w/o Sink tokens | w/ Sink tokens | Overall | w/o Sink tokens | w/ Sink tokens | Overall | w/o Sink tokens | w/ Sink tokens |
| Dynamic | 4 | 1.30 | 1.38 | 0.07 | 0.10 | 0.11 | 0.01 | 0.28 | 0.26 | 0.31 |
| | 3 | 9.00 | 9.57 | 0.50 | 0.68 | 0.72 | 0.08 | 2.03 | 1.78 | 2.30 |
| | 2 | 27.58 | 29.32 | 1.37 | 2.53 | 2.68 | 0.23 | 7.05 | 6.52 | 7.58 |
| | Bits | overall | w/o Sink tokens | - | Overall | w/o Sink tokens | - | Overall | w/o Sink tokens | - |
| Static | 4 | 86.73 | 16.39 | - | 9.75 | 3.10 | - | 9.02 | 5.17 | - |
| | 3 | 89.60 | 20.41 | - | 10.09 | 3.45 | - | 10.24 | 8.26 | - |
| | 2 | 107.36 | 46.14 | - | 11.30 | 6.35 | - | 15.83 | 11.44 | - |

Table 1: The experiments are conducted using round-to-nearest (RTN) integer quantization and the mean squared error (MSE) metric on the LLaMA2-7B model, with the values in the table scaled by a factor of 100. The quantization group size is uniformly set to 16, and global min-max is used for static quantization. Under dynamic quantization, *w/o Sink tokens* refers to MSE for quantization groups that exclude sink tokens, while *w/ Sink tokens* refers to MSE solely for groups containing sink tokens. Under static quantization, *w/o Sink tokens* indicates that sink tokens are excluded during both the calibration and quantization.

**High cosine similarity of QK.** Although the norms of Queries and Keys are small, the cosine similarity between the Queries of non-sink tokens and the Keys of the sink tokens remains high (Gu et al., 2024), resulting in large attention scores, as shown in Figure 5a.

Due to these two mechanisms, a small number of sink tokens exhibit extremely high attention scores but small Values, while the remaining tokens receive lower attention scores, resulting in attention outputs with small values, as shown in Figure 5f. This perspective aligns with previous research (Bondarenko et al., 2023) on extreme outliers in pre-LLM Transformers, but we arrive at this conclusion from a novel angle by analyzing the stabilization of stable outliers during LLM inference. Notably, the QKV suppression mechanism imposes distinct numerical characteristics on the Queries, Keys, and Values of sink tokens, which is the fundamental reason for their sensitivity to quantization. Additional experimental results on QKV suppression and high cosine similarity of QK are provided in Appendix E.

## 4 KV Cache Quantization and Attention Sinks

### 4.1 Impact of Attention Sinks on KV Cache Quantization

Due to QKV suppression, the inclusion of sink tokens within a quantization group can expand the quantization range and exacerbate quantization errors, ultimately leading to further performance degradation. This arises from the trade-off between range and precision. Recent KV cache quantization methods adopt diverse quantization schemes (Liu et al., 2024c; Hooper et al., 2025; Duanmu et al., 2024), yet the impact of attention sinks across these schemes remains largely unexplored. In this section, we conduct an comprehensive study of the impact of attention sinks on widely adopted per-token and per-channel approaches

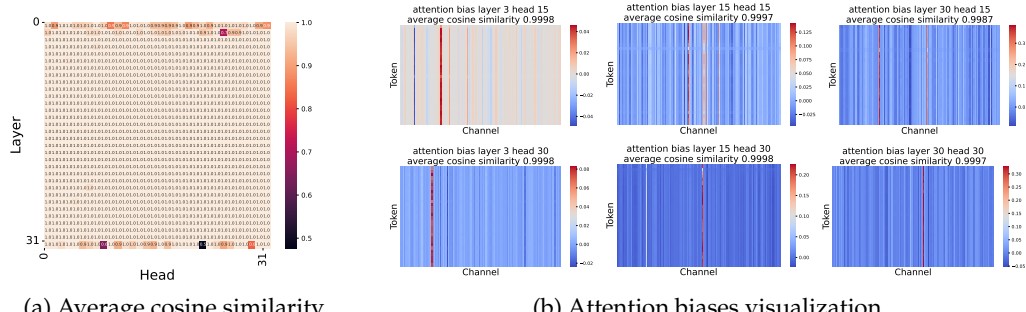

(a) Average cosine similarity          (b) Attention biases visualization

Figure 6: (6a) depicts the average cosine similarity of $\sum_{i \in S} p_i^t v_i$ across all tokens for each head on LLaMA2-7B. (6b) visualizes the attention biases for several example heads.

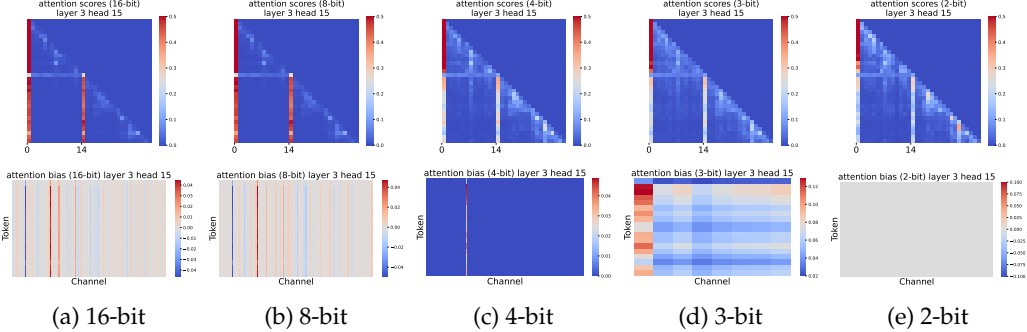

(a) 16-bit        (b) 8-bit        (c) 4-bit        (d) 3-bit        (e) 2-bit

Figure 7: The impact of KV cache quantization on attention scores and attention biases across various bit-widths, employing static per-token quantization with a group size of 16.

under both dynamic and static quantization through quantization error analysis (presented in Table 1). An overview of low-bit quantization is provided in Appendix C.

**Per-token quantization.** For per-token dynamic quantization, since quantization parameters are computed within each group corresponding to a single token, the QKV suppression of sink tokens does not influence other tokens. However, in static quantization, where parameters are calibrated and fixed throughout quantization, the influence of attention sinks propagates across all quantization groups, exacerbating performance degradation. As shown in Table 1, excluding sink tokens during static per-token quantization significantly reduces quantization error, decreasing by up to 81.1% for Keys and 68.2% for Values.

**Per-channel Key quantization.** Based on the observation that the Keys exhibit outliers in specific channels, while Values do not, some research (Hooper et al., 2025; Liu et al., 2024c) employs per-channel quantization for Keys. In per-channel dynamic Key quantization, groups containing sink tokens experience larger quantization errors, while those without remain unaffected. As shown in Table 1, quantization errors increase by 16.3% to 29.2% compared to those without sink tokens. In per-channel static Key quantization, similar to per-token static quantization, excluding the impact of sink tokens leads to a reduction in quantization error of up to 42.7%.

### 4.2 Effect of KV Cache Quantization on Attention Sinks

Several previous research (Sun et al., 2024; Gu et al., 2024; An et al., 2025) suggest that attention sinks and extreme activation outliers introduce implicit attention biases and demonstrate that incorporating explicit learnable biases during training can effectively eliminate outliers. Building on this insight, we first validate the presence of attention biases induced by attention sinks during inference and then analyze the impact of KV cache quantization on these biases. The attention output of token $t$ can be expressed as:

$$\text{Attention}(Q, K, V)^t = \sum_{i \leq t} p_i^t v_i = \sum_{i \notin S} p_i^t v_i + \sum_{i \in S} p_i^t v_i, \tag{9}$$

where $p_i^t$ represents the attention score of the Query token $t$ and the Key token $i$, $S$ denotes the set of sink tokens and $v_i$ denotes the Value of token $i$. As shown in Equation 9, the

presence of attention sinks influences the attention output of each token $t$ through $\sum_{i \in S} p_i^t v_i$. To verify whether this represents attention biases, we calculate the average cosine similarity of $\sum_{i \in S} p_i^t v_i$ across all tokens for each attention head. As shown in Figure 6, we find that for each head, $\sum_{i \in S} p_i^t v_i$ remains highly consistent across all tokens when attention sinks emerge, confirming that it represents the attention biases.

Next, we analyze the impact of KV cache quantization on attention sinks and attention biases. As shown in Figure 7, quantization significantly affects both, with the impact becoming more pronounced as the bit-width decreases. Notably, since the biases introduced by attention sinks persist across all subsequent tokens and may contain global or other crucial information (Darcet et al., 2023), their influence on attention computation remains continuous and significant.

### 4.3 KVSink

Given the profound interplay between attention sinks and KV cache quantization, it is crucial to implement effective preservation mechanisms during quantization. Existing approaches (Hooper et al., 2025; Duanmu et al., 2024) statically preserve the first few tokens (PFN), overlooking the potential presence of attention sinks at other positions. Moreover, relying on attention scores to identify attention sinks is not a practical solution. First, dynamically identifying sink tokens incurs significant overhead. Second, attention computations rely on optimized CUDA kernels, such as FlashAttention (Dao et al., 2022; Dao, 2023), which do not expose intermediate results. To address this challenge, we propose *KVSink*, with an overview illustrated in Figure 8. As discussed in Section 3, both stable outliers and attention sinks manifest on sink tokens. Building on this, KVSink initially identifies the emergence of stable outliers and subsequently uses them as indicators to predict the positions of sink tokens. Notably, identification of outliers

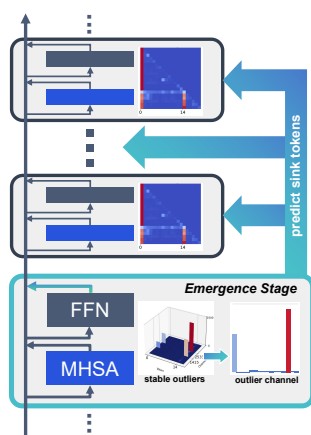

Figure 8: Overview of KVSink.

is both highly efficient and accurate. *(1)* Outliers exhibit extremely large magnitudes and are sparse in occurrence, which enables their efficient detection through a straightforward top-k sorting approach. *(2)* The identification needs to be performed only once during inference at the emergence stage. Additionally, the emergence stage layer can be pre-identified and treated as a static pattern, as it is input-independent (see Appendix D). *(3)* For a given LLM, outliers consistently emerge in specific fixed channels (Sun et al., 2024; An et al., 2025). This allows identification to be restricted to a single pre-identified channel. *(4)* Since the initial input sequence is typically sufficiently long to capture all attention sinks, performing outlier identification only during the prefill phase offers a more computationally efficient approach. The complete algorithm for KVSink is presented in Appendix G.

## 5 Experiments

### 5.1 Experiment Settings

To assess the benefits of KVSink, we first compare it with the existing Preserve-First-N (PFN) solution across various LLMs and KV cache quantization schemes. Additionally, to further highlight the improvements KVSink provides over well-established methods, we conduct experiments using the KVQuant method (Hooper et al., 2025). The efficiency analysis of KVSink is presented in Appendix F.

**Models, tasks, and datasets.** We evaluate KVSink across seven models: LLaMA2-7B/13B/70B, LLaMA2-7B-chat, Mistral-7B, LLaMA3-8B and LLaMA3.1-8B-instruct (Touvron et al., 2023; Chaplot, 2023; Dubey et al., 2024). The evaluation includes PPL tests conducted on the WikiText-2 and C4 datasets (Merity, 2016; Raffel et al., 2020).

**Comparison of KVSink with the PFN strategy.** We compare the PPL reduction achieved by KVSink and PFN using the same number of tokens for preservation, with values set to 0, 5, 10, 15, and 20. The evaluation is conducted based on three basic quantization schemes using

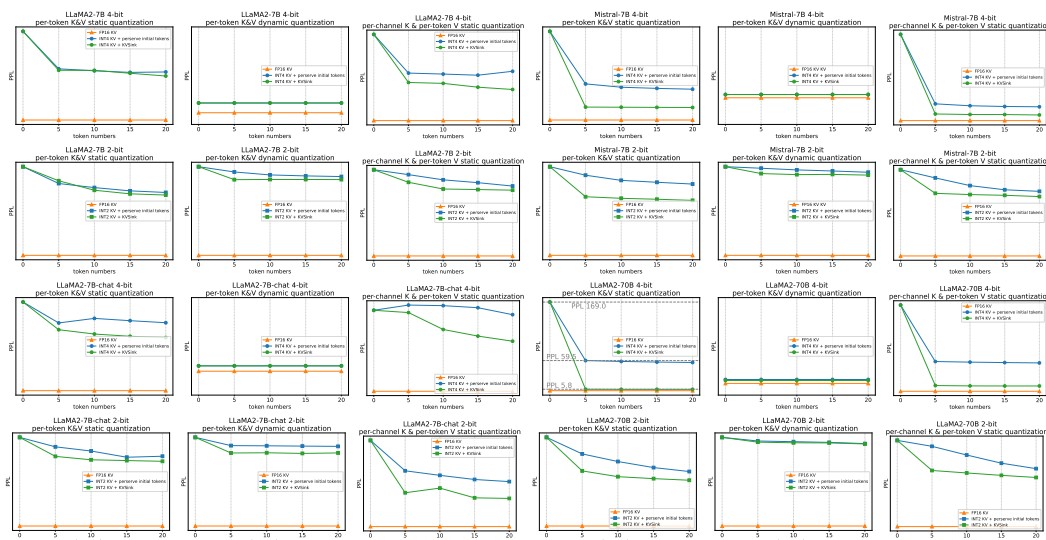

Figure 9: Comparison of KVSink with the PFN strategy. The orange line denotes the PPL of the FP16 KV cache, green indicates the use of KVSink, and the blue line represents the PFN method. A PPL closer to that of FP16 reflects better performance.

| Methods | WikiText-2 PPL ↓ | | | | C4 PPL ↓ | | | | KV Avg. Bits |
|---|---|---|---|---|---|---|---|---|---|
| | LLaMA2 7B | LLaMA2 13B | LLaMA3 8B | LLaMA3.1 8B-Instruct | LLaMA2 7B | LLaMA2 13B | LLaMA3 8B | LLaMA3.1 8B-Instruct | |
| 16-bit | 5.12 | 4.57 | 5.75 | 6.75 | 6.63 | 6.05 | 7.37 | 8.03 | 16 |
| KVQuant | 7.46 | 16.26 | 7.31 | 9.75 | 9.46 | 26.29 | 8.53 | 10.43 | 2-2.02 |
| + *KVSink-5* | 6.52 | 5.10 | 6.77 | 8.83 | 7.93 | 6.47 | 7.74 | 9.57 | |
| KVQuant-0.1% | 5.73 | 5.00 | 6.84 | 8.80 | 7.11 | 6.39 | 8.06 | 9.25 | 2.04-2.06 |
| + *KVSink-5* | 5.60 | 4.85 | 6.56 | 8.25 | 7.10 | 6.28 | 7.87 | 8.80 | |
| KVQuant-0.5% | 5.62 | 4.86 | 6.51 | 7.91 | 7.11 | 6.24 | 7.87 | 8.71 | 2.16-2.19 |
| + *KVSink-5* | 5.54 | 4.76 | 6.34 | 7.69 | 6.95 | 6.22 | 7.81 | 8.65 | |
| KVQuant-1% | 5.53 | 4.80 | 6.37 | 7.58 | 6.94 | 6.23 | 7.82 | 8.53 | 2.32-2.35 |
| + *KVSink-5* | 5.44 | 4.71 | 6.22 | 7.47 | 6.81 | 6.19 | 7.78 | 8.50 | |

Table 2: Application to the KVQuant method. KVSink-5 denotes the use of 5 sink tokens.

RTN INT2/INT4 quantization: per-token Key and Value static, per-token Key and Value dynamic, and per-channel Key with per-token Value static. The quantization group size is uniformly set to 128, and the Wikitext-2 dataset is used for the PPL test.

**Application of KVSink to the KVQuant method.** KVQuant (Hooper et al., 2025) employs a range of advanced techniques for ultra-low-bit KV cache quantization, delivering state-of-the-art performance, including non-uniform quantization, per-vector dense-and-sparse quantization for isolating numerical outliers, and per-channel quantization for Keys along with per-token quantization for Values. However, to preserve attention sinks, KVQuant only excludes the first token during calibration and quantization, a limitation that KVSink aims to enhance. We apply KVSink to the 2-bit KVQuant method with various settings for numerical outliers, including 1%, 0.5%, 0.1%, and no isolation of numerical outliers. The evaluation is performed using PPL tests on the Wikitext-2 and C4 datasets.

## 5.2 Main Results

**Comparison with the PFN strategy.** As shown in Figure 9, KVSink outperforms PFN in almost all cases. For instance, when evaluating LLaMA2-70B with per-token KV static 4-bit quantization, preserving only 5 sink tokens with KVSink results in a PPL reduction of 163.2, with only a marginal increase of 2.5 in PPL compared to the FP16 baseline. In contrast, the PFN strategy exhibits the PPL of 59.5, as it fails to account for sink tokens at other positions. Additionally, the experiments also show that preserving only 5 sink tokens with KVSink is sufficient in most cases.

**Application to the KVQuant method.** As shown in Table 2, integrating KVSink with the well-established KVQuant method provides two key benefits. First, while KVSink introduces only minimal modifications, it consistently improves PPL, with the benefits

becoming more pronounced as fewer numerical outliers are preserved. Second, KVSink reduces the dependence on preserving FP16 numerical outliers. For example, with the 0.1% setting, the performance with KVSink is maintained or even exceeds that of the 0.5%. Notably, reducing the preservation of FP16 outliers leads to more efficient compression.

## 6 Conclusion

In this study, we enhance the understanding of attention sink preservation in KV cache quantization and propose KVSink to improve the existing PFN solution. We elucidate the intrinsic relationships in the cross-layer evolution of different types of extreme activation outliers and highlight the pivotal role of attention sinks during the stabilization stage of stable outliers. We also thoroughly analyze the mutual influence between attention sinks and KV cache quantization. Experimental results demonstrate that KVSink outperforms the existing PFN strategy and improves established KV cache quantization techniques.

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

## A Related Work on Attention Sinks and Extreme Activation Outliers in Transformer-Based Models

Previous studies have revealed the widespread presence of extreme activation outliers in Transformer-based models (Bondarenko et al., 2021), including BERT (Devlin et al., 2019; Kovaleva et al., 2019; Clark et al., 2019), Vision Transformer (ViT) (Dosovitskiy et al., 2020; Bondarenko et al., 2023; Sun et al., 2024; Darcet et al., 2023), and LLM (Sun et al., 2024; Guo et al., 2024; Yang et al., 2024a), with substantial attention concentrating on these outliers, forming attention sinks.

Numerous studies have sought to elucidate this behavior in Transformer models. In pre-LLM research, Bondarenko et al. (2021), as a pioneering study, identified the bottleneck in activation quantization of Transformers caused by extreme outliers and uncovered the intrinsic relationship between attention focus pattern and these outliers. Clark et al. (2019) demonstrates that BERT-like Transformers tend to focus on the special [SEP] token. This behavior effectively acts as a "no-op" for attention heads that are unable to extract the patterns they were trained to detect from the specific passage of text. Bondarenko et al. (2023) found that outliers and attention focus arise as attention heads attempt to learn a "no-op" or a partial update of the residual. In this process, strong outliers emerge due to the softmax function.

StreamLLM (Xiao et al., 2023) has conducted an initial investigation revealing the presence of attention sinks in LLMs. It suggests that LLMs tend to treat initial token as attention sink because the model tends to dump unnecessary attention values to specific tokens. Sun et al. (2024) conducted an in-depth study on massive activations in LLMs and ViTs, demonstrating that these activations give rise to attention sinks and lead to implicit attention biases. Gu et al. (2024) found that attention sinks function more like Key biases, storing extra attention scores that may be non-informative and not contribute to the Value computation. An et al. (2025) categorize three types of outliers—activation outliers, weight outliers, and attention outliers—revealing their intrinsic connections and collective impact on the attention mechanism.

Building on these works, we elucidate the role of attention sinks by analyzing the cross-layer evolution of different types of extreme activation outliers, offering a novel perspective that has not been explored in prior research. Furthermore, our study delves into their interaction with KV cache quantization, offering valuable insights for future investigations.

## B Related Work on KV Cache Quantization

Low-bit quantization reduces the bit-width of the KV cache representation, effectively decreasing its size and thereby mitigating memory usage and access bottlenecks. However, this process inevitably introduces quantization errors, leading to performance degradation. Existing methods explore various approaches to mitigate the impact of quantization errors in KV cache quantization. These methods can generally be classified into two main types based on the selected quantization dimension of the Keys.

KVQuant (Hooper et al., 2025) and KIVI (Liu et al., 2024c) both observe that Keys exhibit outliers in specific channels, while Values do not. Based on this observation, both methods utilize per-channel quantization for Keys to reduce the quantization difficulty. KVQuant employs non-uniform static quantization and incorporates several optimization techniques to address the challenges associated with static quantization. These techniques include pre-RoPE quantization, per-vector dense-and-sparse quantization, and attention-sink-aware quantization, among others. KIVI employs dynamic integer quantization, where KV cache quantization is applied after accumulating a specified number of local tokens.

On the other hand, per-token KV cache quantization often uses token-level mixed-precision quantization to preserve the precision of the KVs of critical tokens, thereby minimizing the loss of essential information. SKVQ (Duanmu et al., 2024), RotateKV (Su et al., 2025a), MiKV (Yang et al., 2024b) and ZipCache (He et al., 2024) are all approaches that focus on per-token mixed-precision quantization. SKVQ introduces clipped dynamic quantization

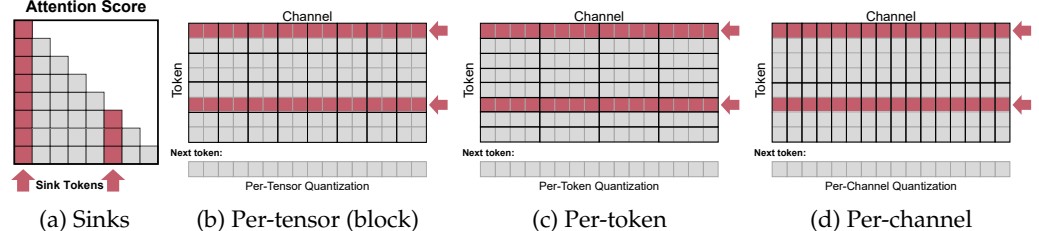

| (a) Sinks | (b) Per-tensor (block) | (c) Per-token | (d) Per-channel |

Figure 10: KV quantization based on different dimensionalities. The tokens highlighted in red represent the Keys or Values of sink tokens.

with channel reordering, preserving high precision for both the initial and most recent tokens. RotateKV utilizes outlier-aware Hadamard-transform-based rotation to reduce the quantization difficulty of the Keys, while preserving high precision for KV pairs associated with attention sink tokens. MiKV assesses token importance using accumulated attention scores, similar to the approach used in H2O (Zhang et al., 2023), and subsequently employs relatively higher bit-widths to preserve the KVs of important tokens. ZipCache utilizes normalized attention scores to more accurately identify salient tokens and incorporates an efficient approximation of the saliency metric.

Although most of these methods preserve higher precision for the KVs of sink tokens or tokens with high attention scores, they lack a clear explanation of the underlying principles. In contrast, our work offers a deeper understanding of the interaction between attention sinks and KV quantization, while also enhancing the existing Preserve-First-N solution.

## C  Overview on Low-Bit Quantization

In this section, we introduce the quantization process using the commonly employed asymmetric integer quantization technique. The $n$-bit asymmetric integer quantization and dequantization processes, where $n \in \mathbb{N}$, can be expressed as:

$$Q(X) = clamp \left( \left\lfloor \frac{X}{scale} \right\rceil + zero, 0, 2^n - 1 \right), \tag{10}$$

$$X' = scale \cdot (Q(X) - zero), \tag{11}$$

$$scale = \frac{clipped\_max(X) - clipped\_min(X)}{2^n - 1}, \tag{12}$$

$$zero = - \left\lfloor \frac{clipped\_min(X)}{scale} \right\rceil, \tag{13}$$

where $\lfloor \cdot \rceil$ indicates round operation. $Q(X)$ and $X'$ denote the quantized and dequantized values of $X$, respectively. The *clamp* operation ensures that the values are constrained within the specified range. The operations $clipped\_max(X)$ and $clipped\_min(X)$ denote the operations that truncate the maximum and minimum values of $X$.

**Quantization dimension.** Quantization can be applied along various dimensions, with common approaches in KV cache quantization including per-tensor, per-token, and per-channel methods, as briefly shown in Figure 10. Per-tensor quantization utilizes a single set of quantization parameters for the entire tensor (or block). Since LLMs perform autoregressive inference, per-token KV quantization has become a widely adopted approach (Duanmu et al., 2024; Su et al., 2025b; He et al., 2024). Recent studies (Liu et al., 2024c; Hooper et al., 2025) have identified outliers along the channel dimension in the Keys, with per-channel quantization shown to effectively mitigate quantization errors. Furthermore, quantization granularity can be improved by defining smaller groupings along the quantization dimension, which reduces quantization errors but introduces additional overhead due to the increased number of quantization parameters.

**Dynamic and static quantization.** Dynamic quantization adjusts the quantization parameters during inference, offering greater flexibility but potentially leading to less efficient performance compared to static quantization due to the additional computational demands

| Model | Total Layers | Emergence Stage Layer of Stable Outliers | Hidden Size | Outlier Channels of Stable Outliers |
|---|---|---|---|---|
| LLaMA2-7B | 32 | 1 | 4096 | 2533, 1415 |
| LLaMA2-13B | 40 | 3 | 5120 | 4743, 2100 |
| Mistral-7B | 32 | 1 | 4096 | 2070, 3398 |
| LLaMA3-8B | 32 | 1 | 4096 | 788, 1384, 4062 |
| LLaMA3.1-8B-instruct | 32 | 1 | 4096 | 788, 1384, 4062 |
| LLaMA3.2-1B | 16 | 1 | 2048 | 400, 698, 2029, 1159 |
| LLaMA3.2-3B | 28 | 1 | 3072 | 588, 1016, 3046, 1731 |

Table 3: Emergence stage and outlier channels of stable outliers for several models.

of online adjustment. In contrast, static quantization involves estimating the range by passing a few batches of calibration data through the model prior to inference. This approach enhances inference efficiency, as the quantization parameters are pre-calculated and remain fixed during inference. However, it may result in higher quantization errors due to the inability to adapt to varying input distributions during runtime.

**Impact of attention sinks on KV cache quantization.** As shown in Figure 10, the abnormal value characteristics of sink tokens resulting from QKV suppression can significantly impact quantization when sink tokens are included in quantization groups or when quantization parameters calibrated using sink tokens are applied. The impact of sink tokens varies significantly across different quantization schemes, as discussed in Section 4.

# D  Additional Experimental Results on Cross-Layer Evolution of Extreme Activation Outliers

In this section, we present additional experiments on the cross-layer evolution of different types of extreme activation outliers across various inputs and models. We conduct experiments using two distinct prompts. These prompts are:

•prompt 1: "The following are multiple choice questions (with answers) about machine learning.\n \n A 6-sided die is rolled 15 times and the results are: side 1 comes up 0 times;"

•prompt 2: "Summer is warm.\n Winter is cold.\n Spring is mild.\n Autumn is crisp.\n The sun rises early in the summer.\n The days are short in the winter.\n "

As shown in Figures 12, 13, and 17, different inputs do not affect the layers at each stage or the channels where stable outliers emerge. Therefore, they can be used as pre-identified static features during inference. Table 3 shows the emergence stage layers and outlier channels for several models used in KVSink.

We then validate the behavior of each type of outlier at each stage on additional models, including models of different sizes, fine-tuned models, and models using Grouped-Query-Attention (GQA) (Ainslie et al., 2023). As shown in Figures 17, 14, and 16, validation across additional models confirms our findings that stable outliers, driven by outliers in $X_{d\_in}^l$ and $X_{d\_out}^l$, undergo a key process of emergence, stabilization, and dissipation. Building on this, the behavior varies slightly across models. For instance, in the LLaMA3 series models, the dissipation stage occurs in the final layer, with no distinct final stage. In the Mistral models, the dissipation stage spans multiple layers, whereas for the other models in our experiments, this stage is confined to a single layer. These minor differences do not impact KVSink's ability to predict sink tokens during the emergence stage.

# E  Additional Experimental Results on QKV Suppression and High Cosine Similarity of QK

In this section, we present additional experimental results on QKV suppression and the high cosine similarity of QK. First, as shown in Figures 18, 19, 20, and 21, QKV suppression persists across layers when attention sinks occur for different inputs (as detailed in Section 4). Second, as shown in Figures 18, 20, 22, and 23, QKV suppression persists across different models. A notable observation is the occurrence of unusually large norms in the Queries and Values of sink tokens in the final layer, a phenomenon that has not been fully explored

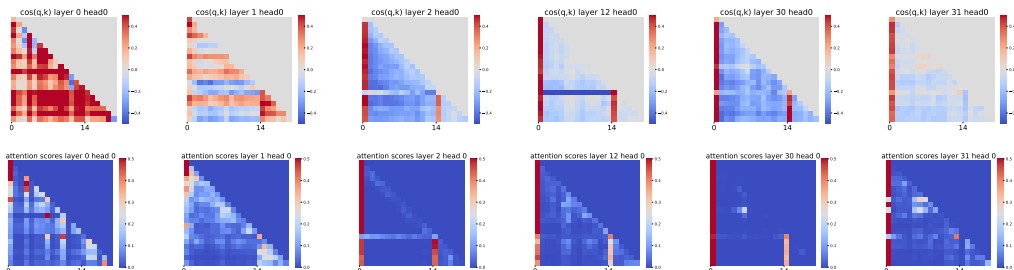

Figure 11: Additional visualizations of the attention scores and the high cosine similarity of QK in LLaMA2-7B.

| Model | LLaMA2-7B | | | | LLaMA2-13B | | | | LLaMA2-70B | | | |
|---|---|---|---|---|---|---|---|---|---|---|---|---|
| | prefill time | | KV Cache memory | | prefill time | | KV Cache memory | | prefill time | | kv cache memory | |
| | w/o KVSink | + KVSink | w/o KVSink | + KVSink | w/o KVSink | + KVSink | w/o KVSink | + KVSink | w/o KVSink | + KVSink | w/o KVSink | + KVSink |
| KVSink-1 | 676.78 | + 0.04 | 256 | + 0.5 | 1090.70 | + 0.04 | 400 | + 0.78 | 4481.05 | + 0.04 | 160 | + 0.31 |
| KVSink-5 | | + 0.04 | | + 2.5 | | + 0.05 | | + 3.91 | | + 0.05 | | + 1.56 |
| KVSink-20 | | + 0.05 | | + 10 | | + 0.05 | | + 15.63 | | + 0.05 | | + 6.25 |

| Model | Mistral-7B | | | | LLaMA3-8B | | | | LLaMA3.2-3B | | | |
|---|---|---|---|---|---|---|---|---|---|---|---|---|
| | prefill time | | KV cache memory | | prefill time | | KV cache memory | | prefill time | | KV cache memory | |
| | w/o KVSink | + KVSink | w/o KVSink | + KVSink | w/o KVSink | + KVSink | w/o KVSink | + KVSink | w/o KVSink | + KVSink | w/o KVSink | + KVSink |
| KVSink-1 | 686.32 | + 0.04 | 64 | + 0.13 | 707.95 | + 0.04 | 64 | + 0.13 | 430.74 | + 0.04 | 56 | + 0.11 |
| KVSink-5 | | + 0.05 | | + 0.63 | | + 0.05 | | + 0.63 | | + 0.05 | | + 0.55 |
| KVSink-20 | | + 0.05 | | + 2.5 | | +0.05 | | + 2.5 | | + 0.05 | | + 2.19 |

Table 4: Efficiency analysis of KVSink. KVSink-N indicates the preservation of N tokens. Time is reported in milliseconds (ms) and memory in megabytes (MB).

in previous research. We hypothesize that this may suggest inherent differences among various sink tokens, and it could be explored as part of future work.

In Figure 11, we present additional visualizations of the attention scores and the high cosine similarity of QK in LLaMA2-7B, confirming the correlation between these two factors.

# F Efficiency Analysis of KVSink

This section presents experimental evaluation and analysis of KVSink's efficiency. We conduct experiments on multiple models using a 4 × A100 (80GB) setup and evaluate performance on the Wikitext-2 dataset, with input sequences segmented into 4K tokens. For time efficiency, we measure the average prefill latency and the time required for KVSink's outlier identification operation under various configurations of sink token quantities. For memory efficiency, we theoretically estimate the original KV cache memory consumption under 2-bit quantization and quantify the additional memory overhead introduced by KVSink's preservation mechanism.

As shown in Table 4, leveraging our enhanced understanding of the attention sinks mechanism, KVSink involves only minimal computations, can be efficiently implemented using PyTorch, and exhibits a negligible impact on time efficiency. The impact on memory efficiency is also minimal. As the context length increases, this impact could diminish further, as the number of preserved tokens remains fixed.

# G KVSink Algorithm

The prefill phase with KVsink is illustrated in Algorithm 1. For clarity, the multi-head mechanism is omitted in the algorithm. Note that if static quantization is applied, sink tokens should also be excluded during quantization parameter calibration.

---

**Algorithm 1** Prefill Phase with KVSink

---

1: **Parameters:** Number of decoder layers: $L$, Emergence stage layer: $l_E$, Hidden size: $d$, Outlier channel: $c$, Number of token length: $n$, Number of tokens for preservation: $k$, Weights: $W_Q, W_K, W_V, W_O$.
2: **Input:** Input to decoder 0: $H^0 \in \mathbb{R}^{n \times d}$.
3: **Output:** Output of decoder $L$: $H^L \in \mathbb{R}^{n \times d}$.
4: **Initialize:** $S_{\text{outliers}} = \varnothing$, $S_{\text{sink}} = \varnothing$
5: **for** $l = 1$ **to** $L$ **do**
6: $\quad H^l \leftarrow \text{LayerNorm}_{mhsa}^l(H^{l-1})$
7: $\quad Q^l \leftarrow H^l \cdot W_Q, K^l \leftarrow H^l \cdot W_K, V^l \leftarrow H^l \cdot W_V$
8: $\quad K^l[\text{token} \notin S_{\text{sink}}] \leftarrow \text{quantize}((K^l[\text{token} \notin S_{\text{sink}}])$
9: $\quad V^l[\text{token} \notin S_{\text{sink}}] \leftarrow \text{quantize}(V^l[\text{token} \notin S_{\text{sink}}])$
10: $\quad K^l \leftarrow K_{\text{quant}}^l, V^l \leftarrow V_{\text{quant}}^l$
11: $\quad H^l \leftarrow \text{Attention}(Q^l, K_{\text{quant}}^l, V_{\text{quant}}^l)$
12: $\quad H^l \leftarrow H^l \cdot W_O + H^{l-1}$
13: $\quad H^l \leftarrow \text{FFN}(\text{LayerNorm}_{ffn}^l(H^l)) + H^l$
14: $\quad$ **if** $l = l_E$ **then**
15: $\qquad S_{\text{outliers}} \leftarrow \left\{ (i, c) \mid |H_{i,c}^l| \in \text{Top-k}(|H_{i,c}^l|) \right\}$
16: $\qquad S_{\text{sink}} \leftarrow \{ i \mid (i, c) \in S_{\text{outliers}} \}$
17: $\quad$ **end if**
18: **end for**
19: **return** $H^L$

---

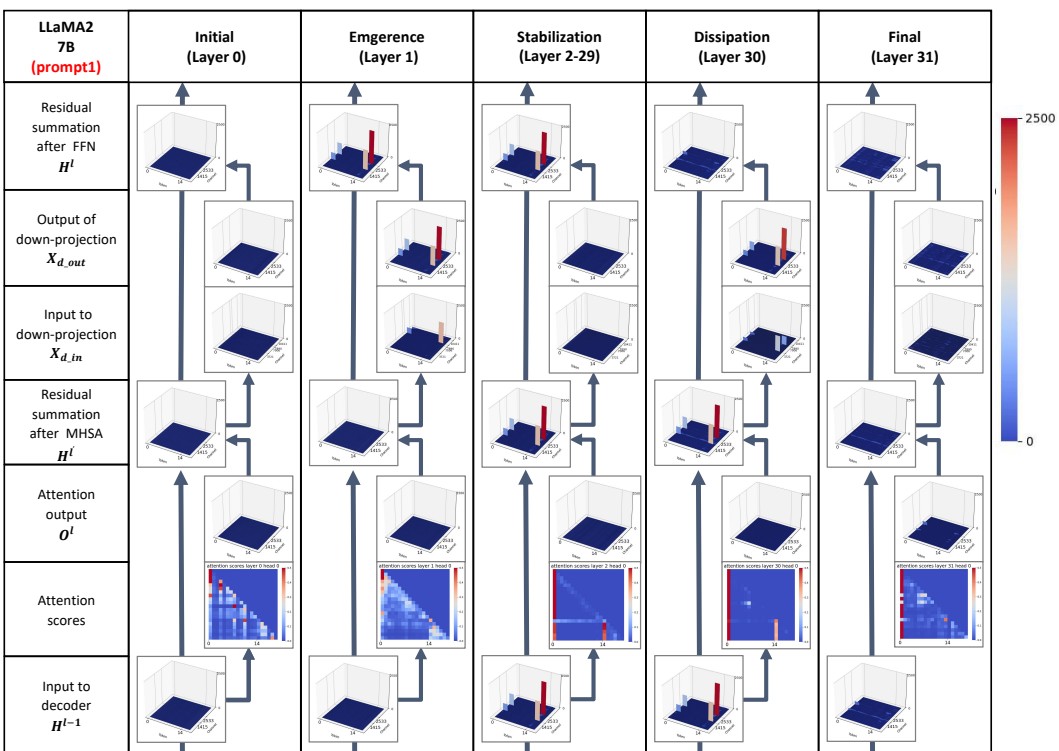

Figure 12: Visualizations of the cross-layer evolution of extreme activation outliers in LLaMA2-7B with Prompt 1.

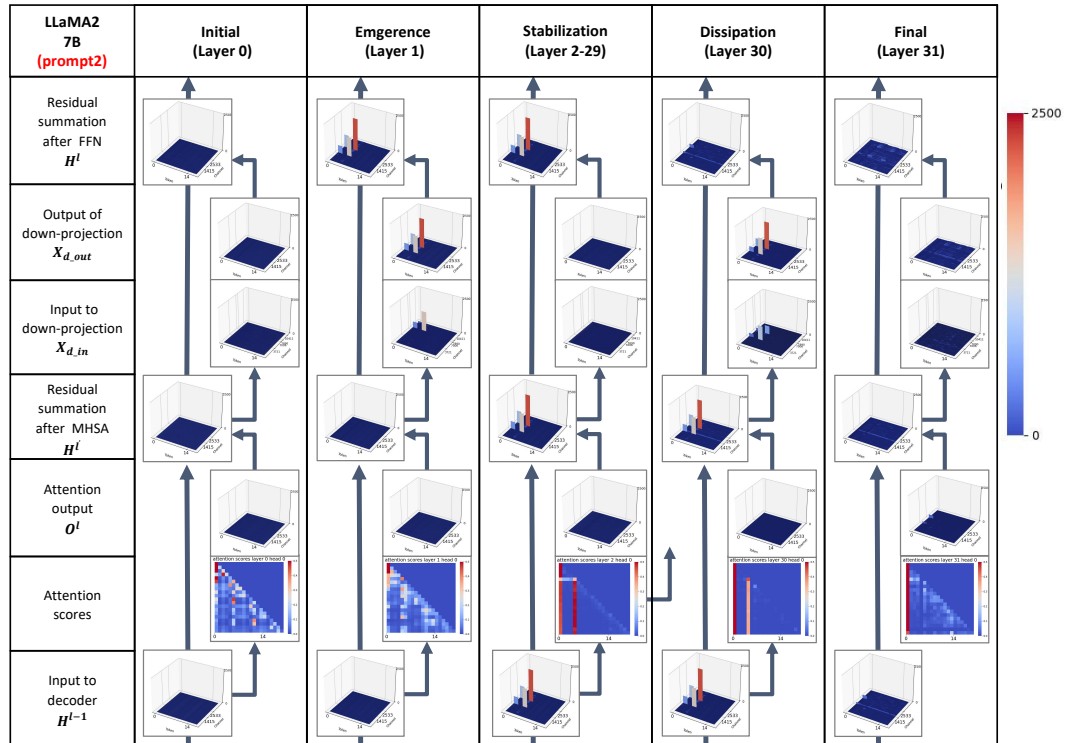

Figure 13: Visualizations of the cross-layer evolution of extreme activation outliers in LLaMA2-7B with Prompt 2.

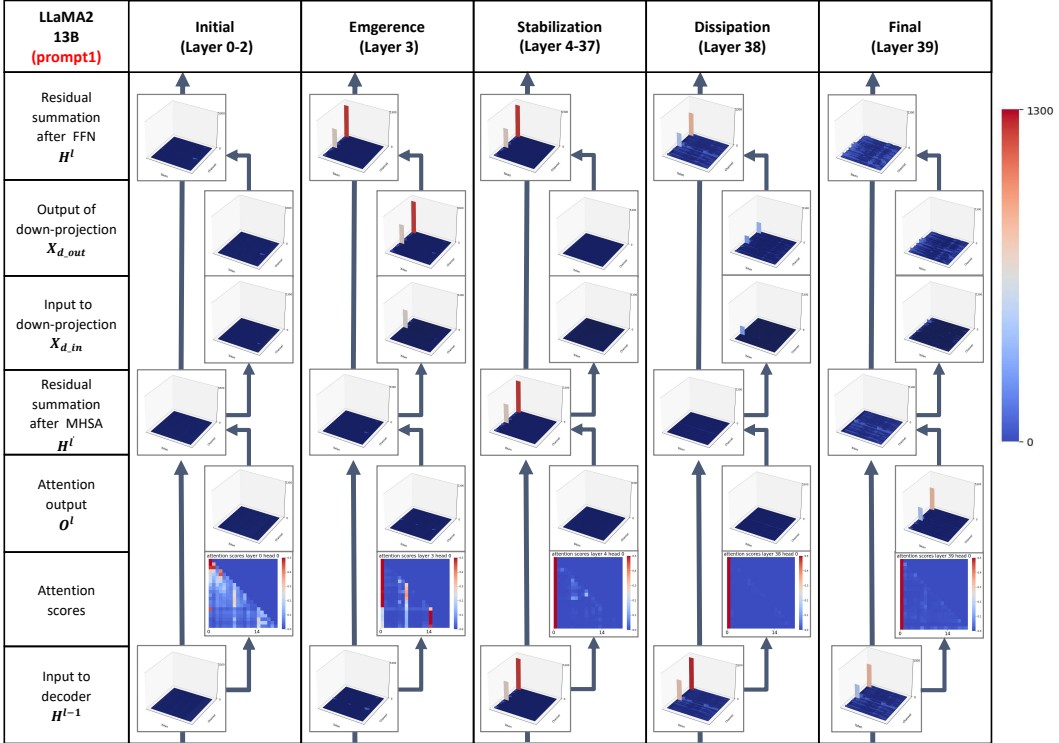

Figure 14: Visualizations of the cross-layer evolution of extreme activation outliers in LLaMA2-13B with Prompt 1.

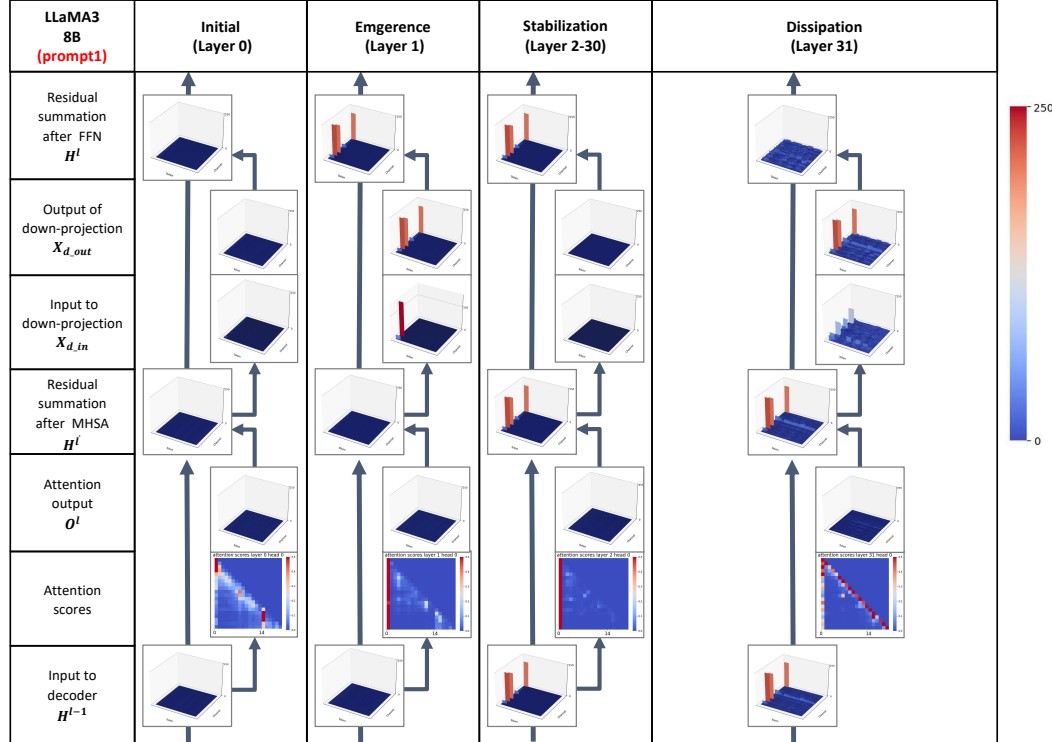

Figure 15: Visualizations of the cross-layer evolution of extreme activation outliers in LLaMA3-8B with Prompt 1.

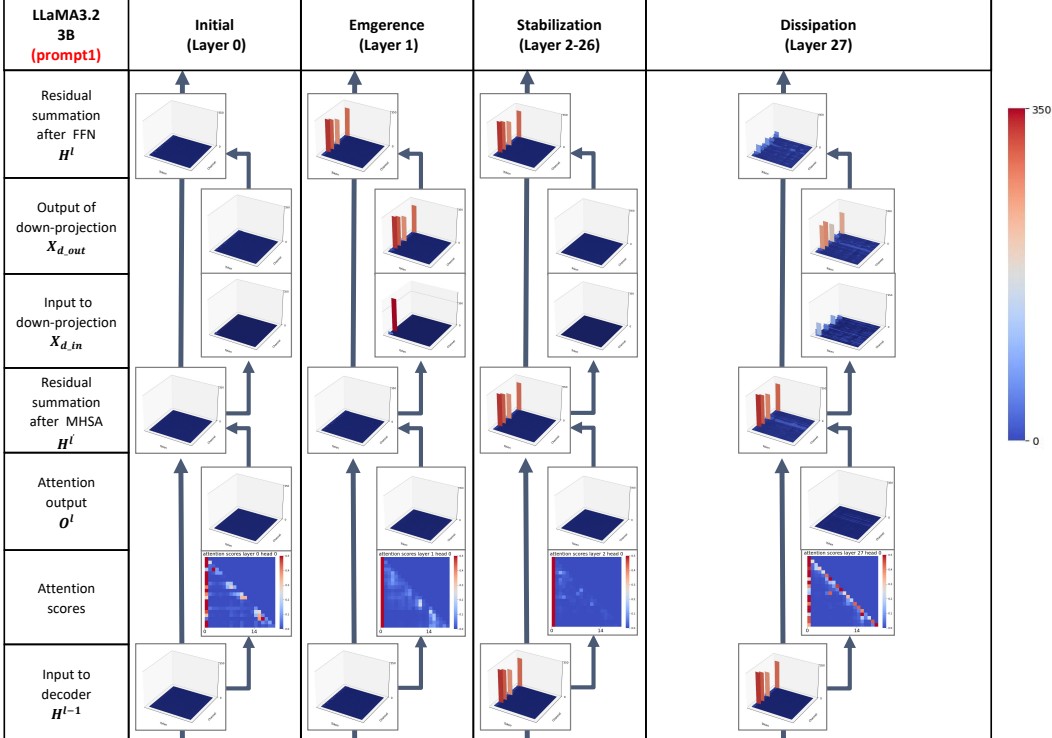

Figure 16: Visualizations of the cross-layer evolution of extreme activation outliers in LLaMA3.2-3B with Prompt 1.

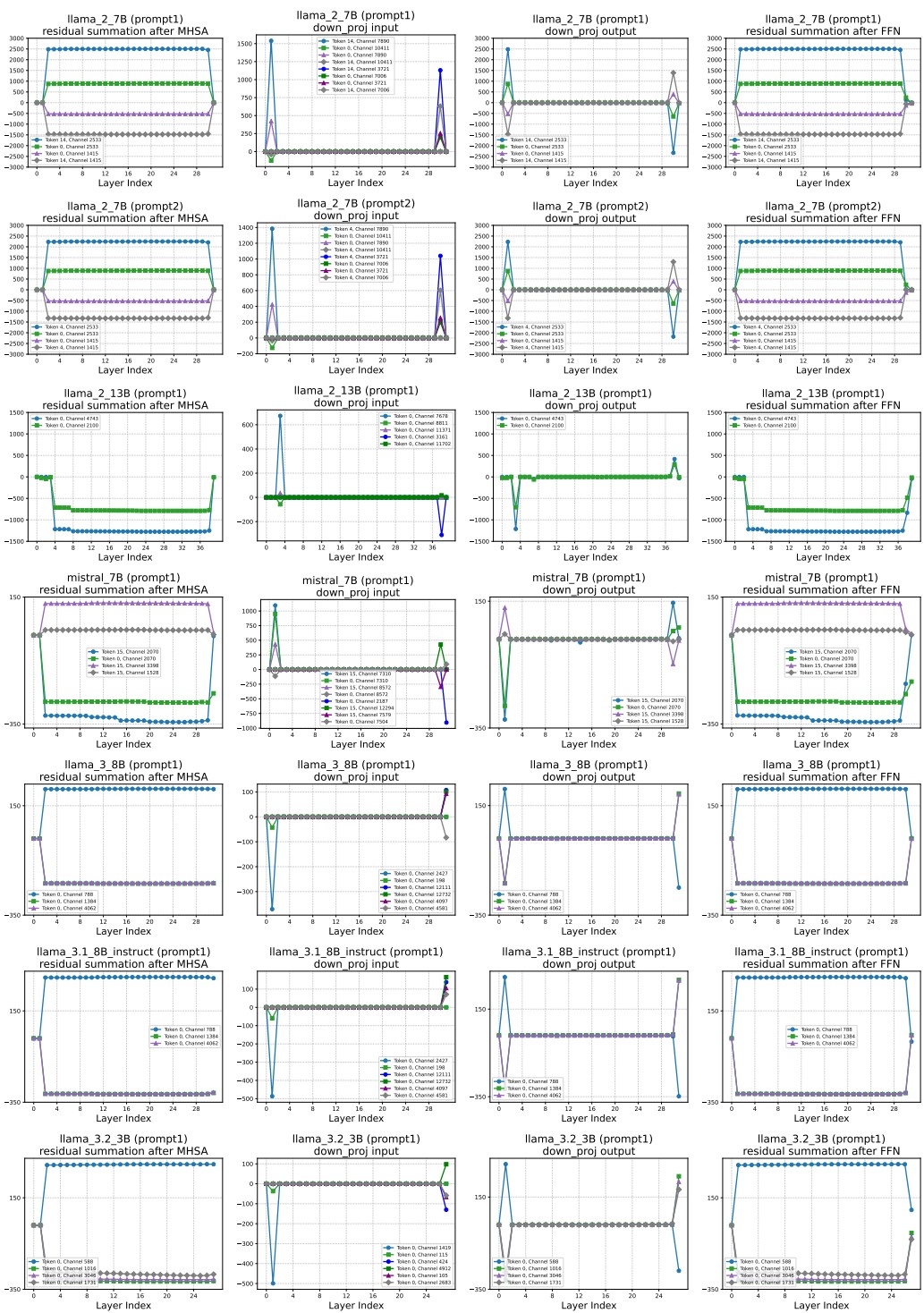

Figure 17: Distribution of extreme activation outliers across decoder layers in multiple models.

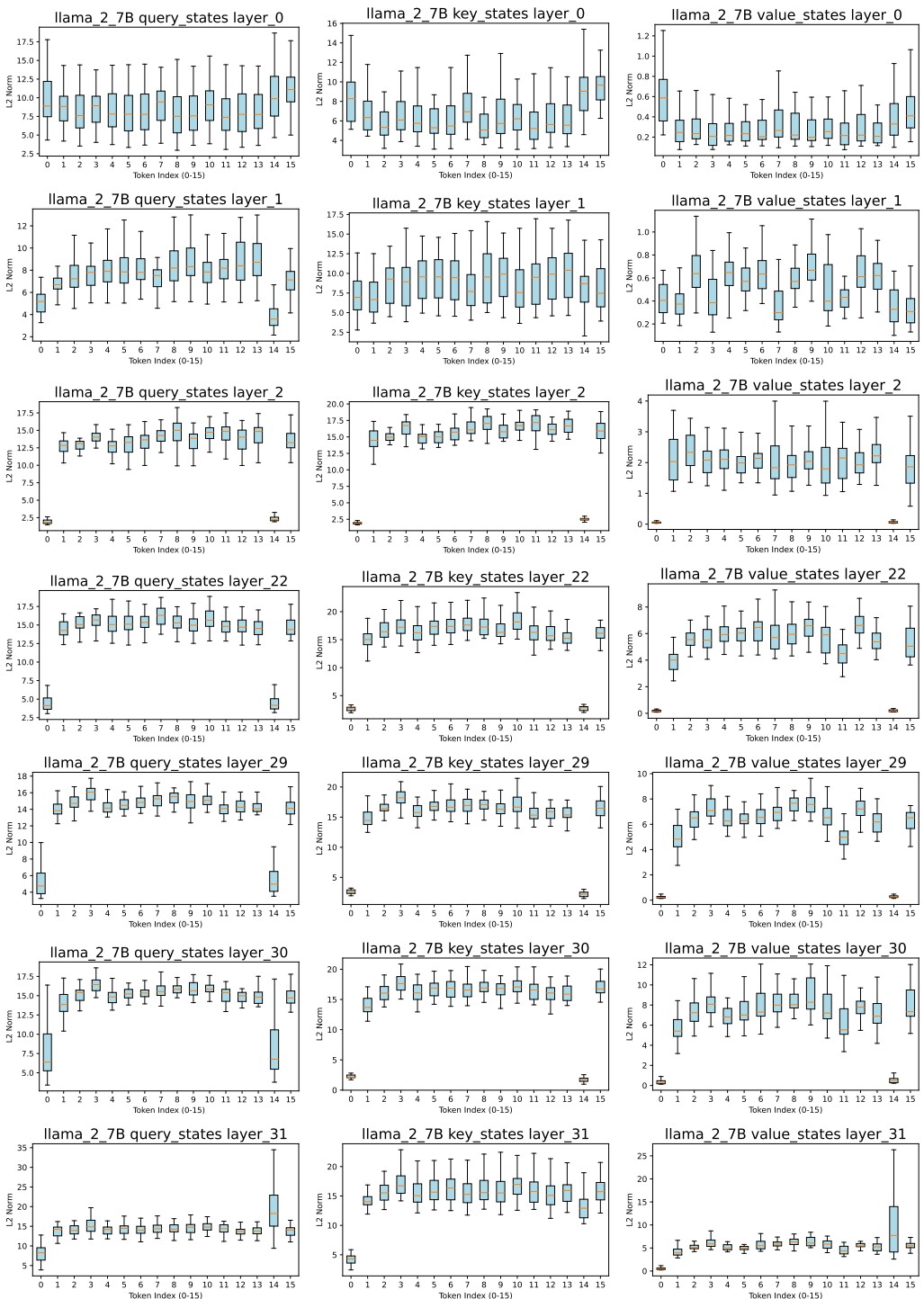

Figure 18: L2 norm distributions of Queries, Keys, and Values for LLaMA2-7B using Prompt 1, with attention sinks occurring in layers beyond layer 0 and 1, at tokens 0 and 14.

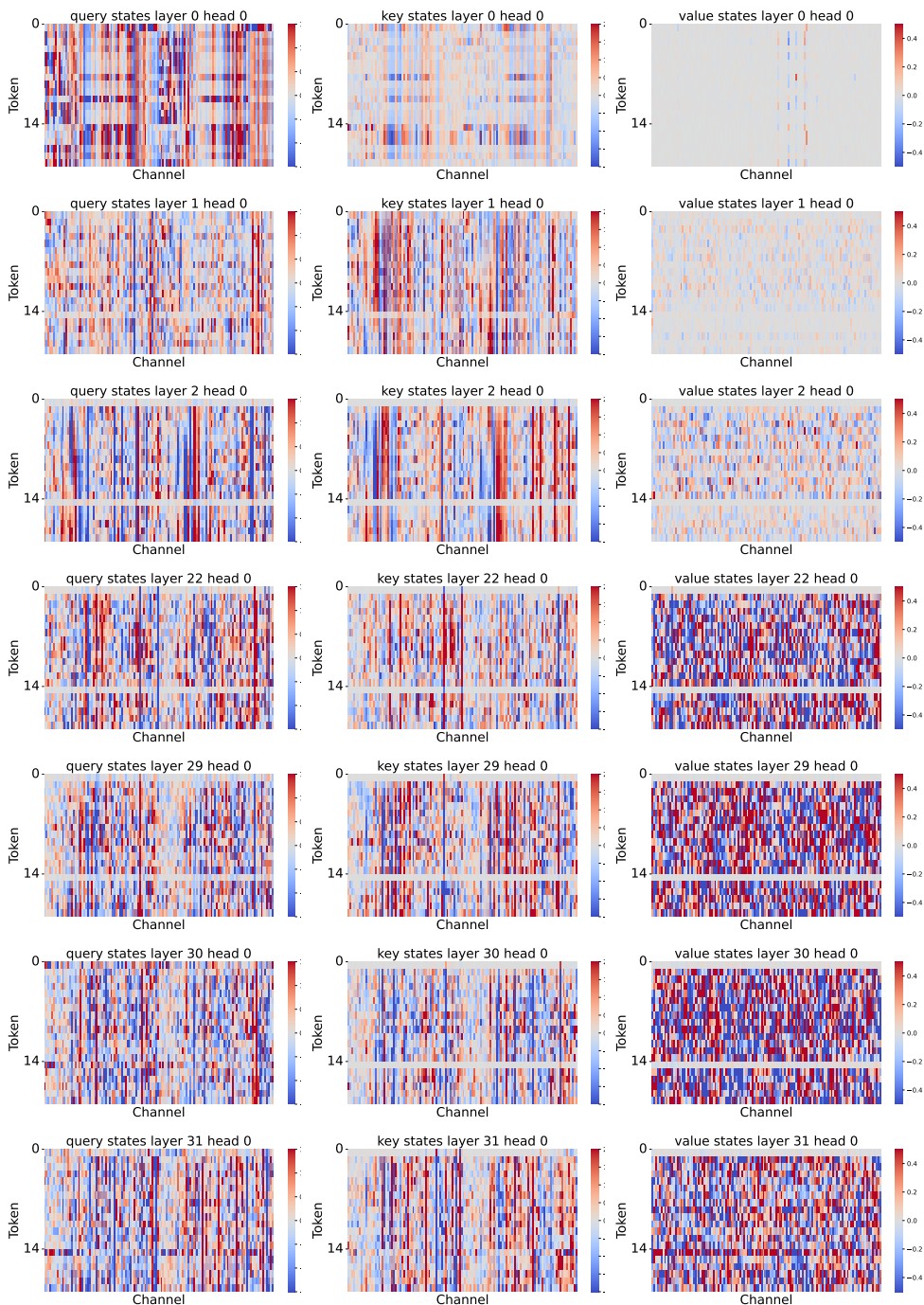

Figure 19: Queries, Keys, and Values for LLaMA2-7B using Prompt 1,with attention sinks occurring in layers beyond layer 0 and 1, at tokens 0 and 14.

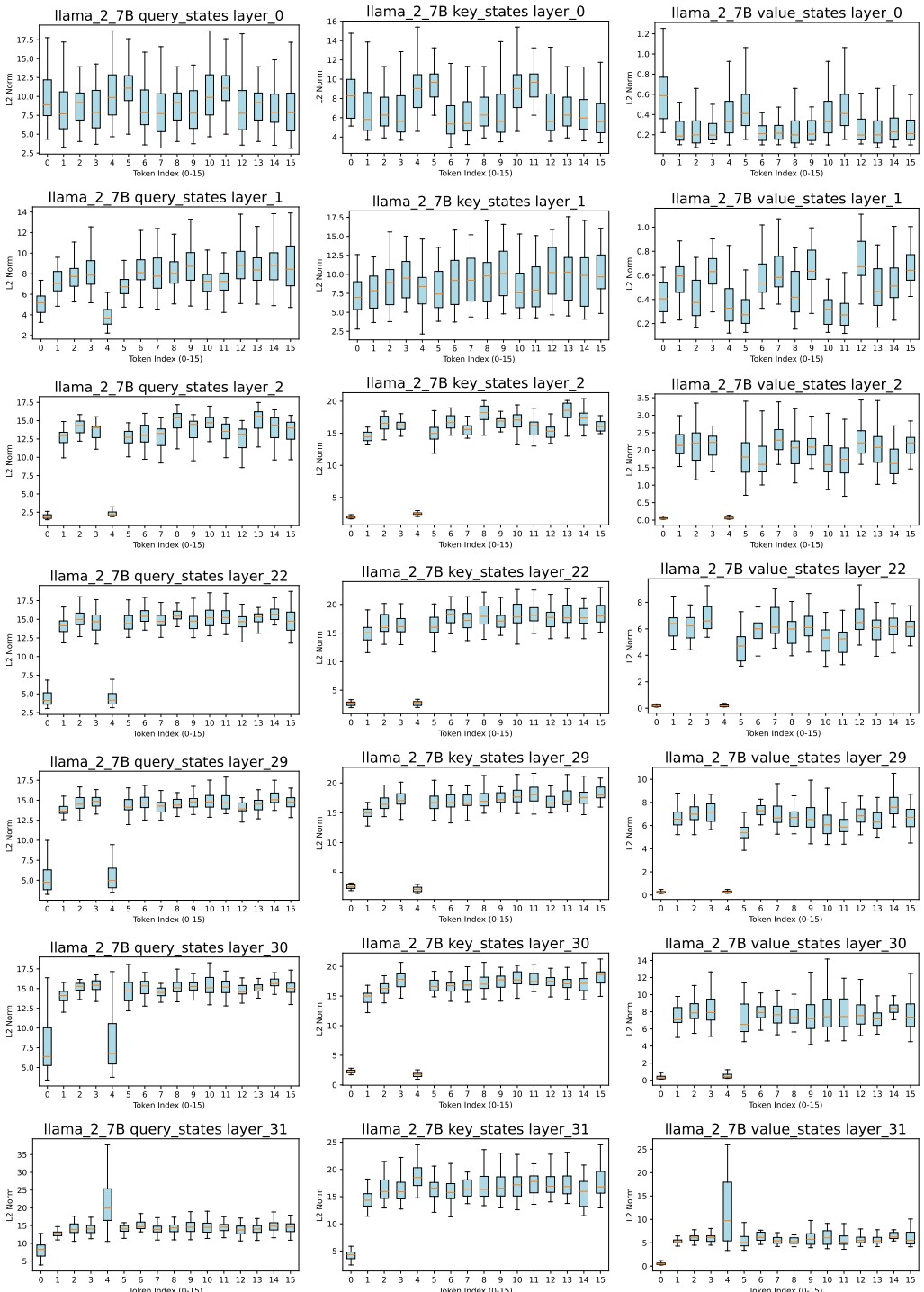

Figure 20: L2 norm distributions of Queries, Keys, and Values for LLaMA2-7B using Prompt 2, with attention sinks occurring in layers beyond layer 0 and 1, at tokens 0 and 4.

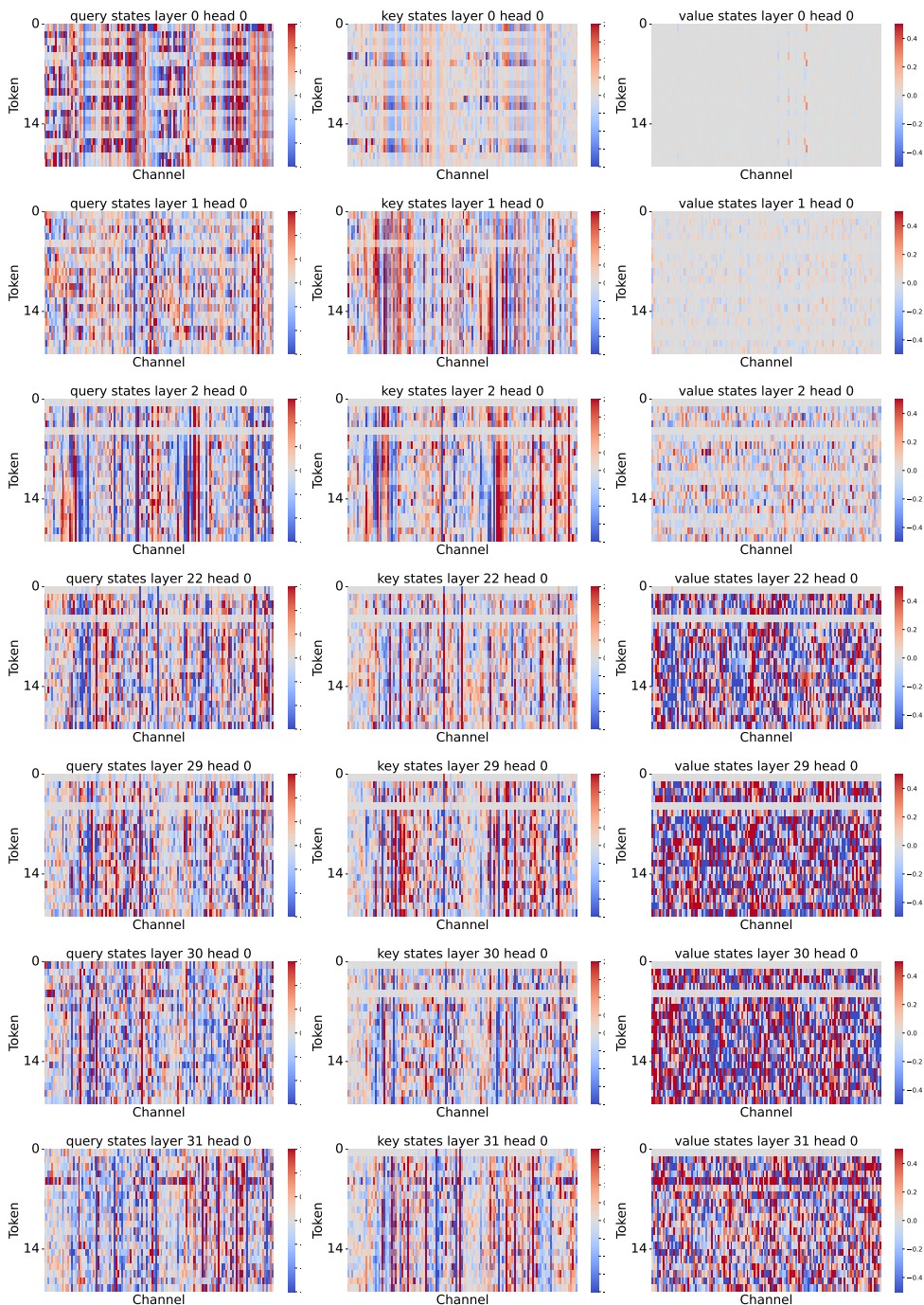

Figure 21: Queries, Keys, and Values for LLaMA2-7B using Prompt 2, with attention sinks occurring in layers beyond layer 0 and 1, at tokens 0 and 4.

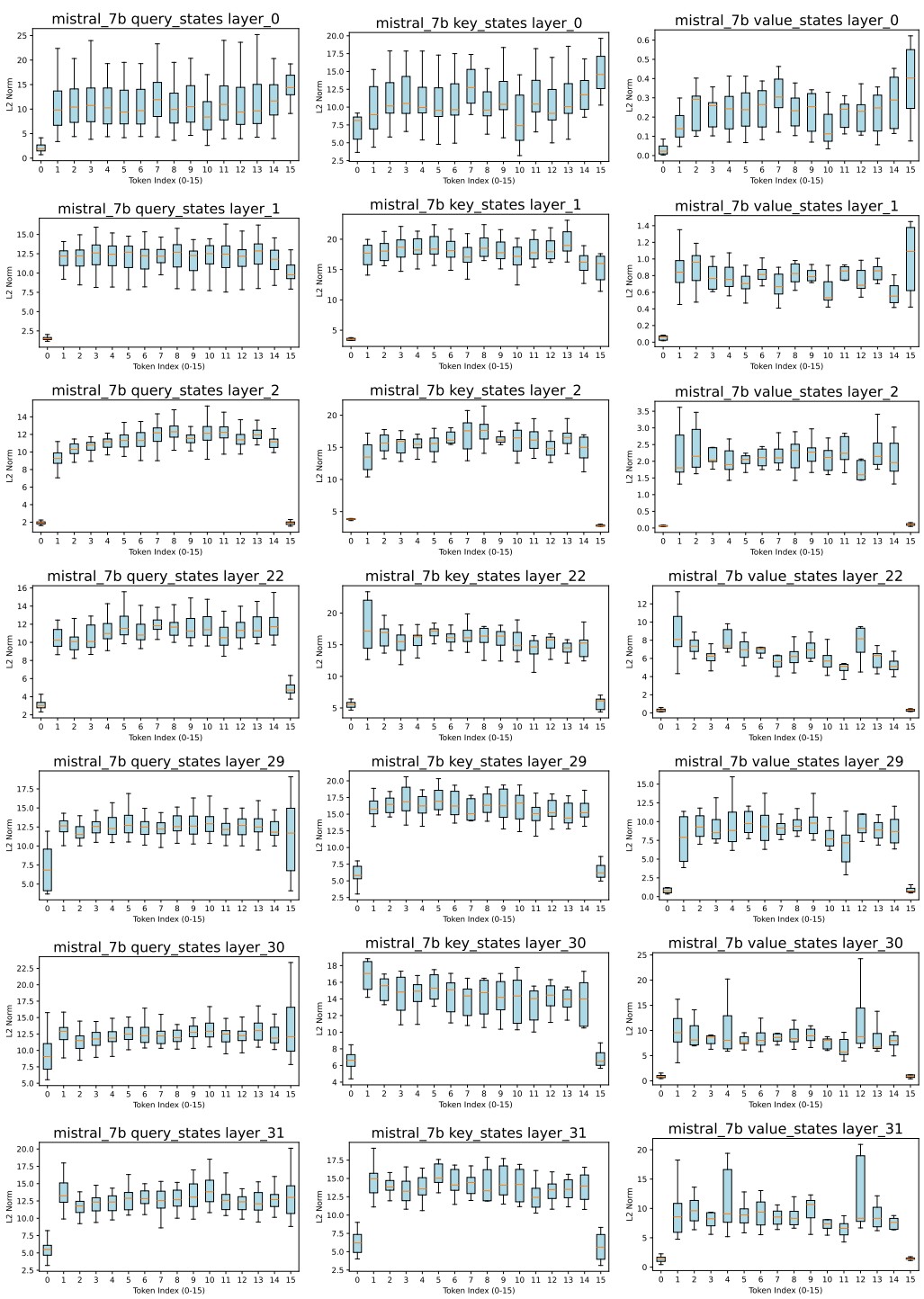

Figure 22: L2 norm distributions of Queries, Keys, and Values for Mistral-7B using Prompt 1, with attention sinks occurring in layers beyond layer 0 and 1, at tokens 0 and 15.

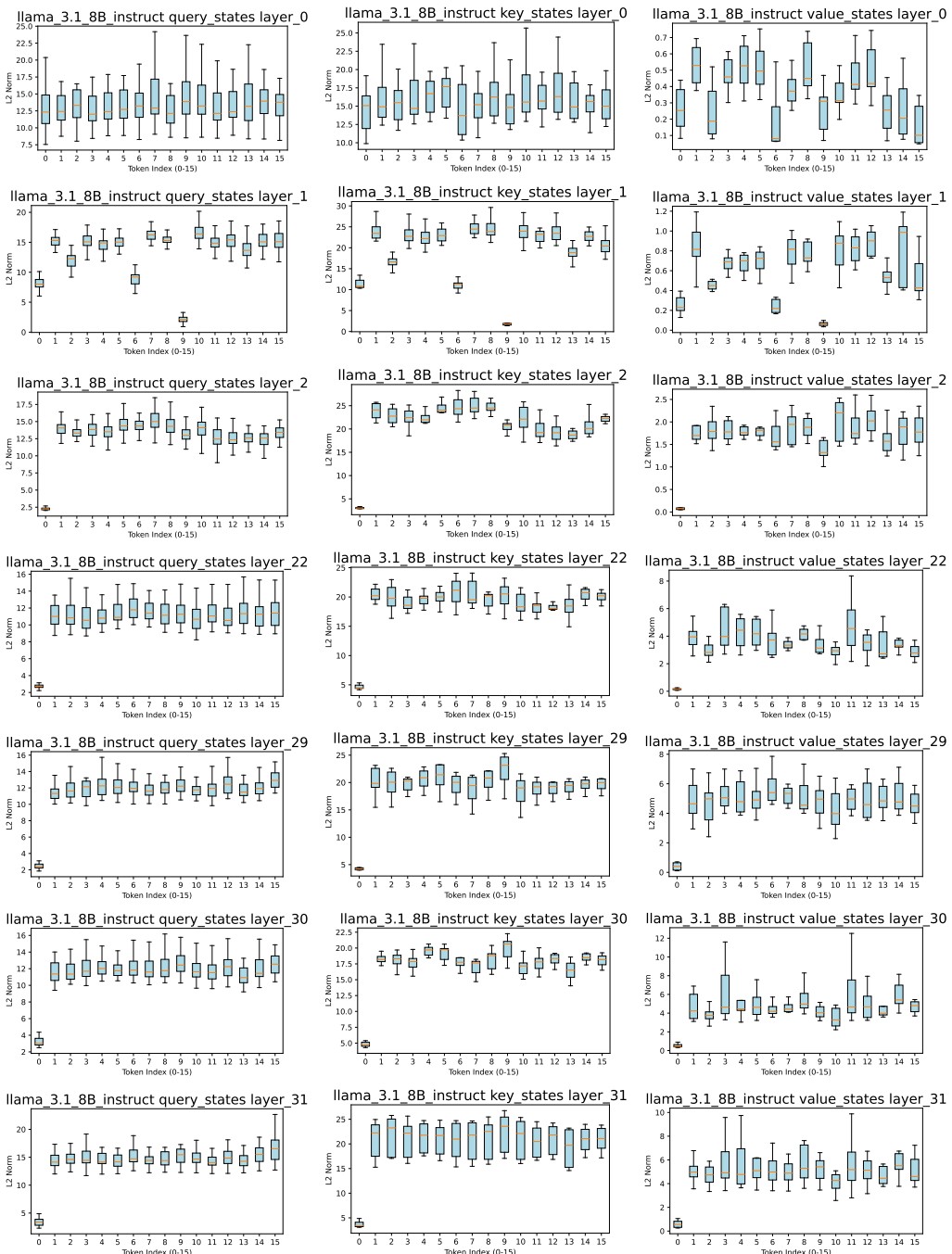

Figure 23: L2 norm distributions of Queries, Keys, and Values for LLaMA3.1-8B-instruct using Prompt 1, with attention sinks occurring in layers beyond layer 0 and 1, at token 0.

