# OpenReview forum: "KVSink: Understanding and Enhancing the Preservation of Attention Sinks in KV Cache Quantization for LLMs"
_colmweb.org/COLM/2025/Conference — COLM 2025_

### Official Review · Reviewer_itTf · 2025-05-06

**Rating:** 6
**Confidence:** 3
**Ethics Flag:** 1

**Summary:**

KVSink enhances KV cache quantization by focusing on the preservation of attention sinks, thereby maintaining the model's ability to attend to initial tokens effectively. This approach strikes a balance between reducing memory usage and preserving model performance, making it particularly beneficial for applications requiring long-context inference.

**Questions To Authors:**

- How well does KVSink generalize to tasks beyond text generation, such as text classification, summarization, or question answering?
- How does it compare with extreme quantization methods like INT4 or mixed-precision approaches?
- Have you considered dynamic sink selection where the attention sinks could adapt based on the input sequence or the task?
- Can you explain any limitations when quantizing the KV tensors? Are there any particular patterns or edge cases where KVSink doesn't perform as well?

**Reasons To Accept:**

KVSink preserves the precision of the KV cache associated with attention sinks selectively. It maintains higher precision for the KV pairs corresponding to the initial tokens, and ensures that the model retains the necessary information to focus on these tokens during inference. This selective preservation mitigates the negative impact of quantization on the model's performance, particularly in tasks where early tokens are crucial.​

**Reasons To Reject:**

- While KVSink selectively retains full precision for the attention sinks, it still quantizes the majority of the KV cache. If important information lies outside the preserved sinks (e.g., middle tokens in long documents), the model may still suffer from precision loss. This can hurt performance on tasks, where important context is spread throughout the sequence, such as summarization or reasoning across long documents.
- Reliance on early tokens consistently attracting attention across layers and tasks, which isn't universally true. Some tasks or prompts may shift attention patterns, in which later tokens can become important depending on the question or query.
- The method is mostly heuristic, e.g., retaining N tokens at the start of the sequence while quantizing the rest.

---

> ### Author Response · Authors · 2025-05-30
>
> We sincerely appreciate the time and effort that Reviewer itTf devoted to evaluating our work.
> The thoughtful and constructive feedback will be very helpful to our future research.
> Below, we will make every effort to address the concerns raised and clarify any potential misunderstandings.
>
> **Response to Reasons to Reject:**
> 1. The concern that quantizing important information tokens at other positions may degrade task performance is indeed valid.
> However, the primary objective of our work is to investigate the underlying principles of how the widely observed attention sink (originally identified by the pioneering work [1]) affect KV quantization, and to provide a plug-and-play enhancement to existing methods.
> Our analysis shows that protecting attention sink tokens is driven by their significant influence on quantization error and implicit attention bias, rather than by the importance of their semantic information.
> 2. We appreciate the concern regarding shifts in attention patterns.
> However, based on existing studies and our observations, attention sinks may represent a special and intriguing exception.
> A notable characteristic of attention sinks is that models consistently disproportionately focus on certain weakly semantic tokens at the beginning of the sequence [1].
> For cross-layer attention patterns, studies such as [1][2][3] have demonstrated that attention sinks consistently appear in attention heads across nearly all layers, regardless of the model input, except for the very first few layers.
> With respect to different models, research [2][3] has observed attention sinks in widely-used open-source models such as Llama, Mistral, Phi, GPT-2.
> 3. Retaining only the first n tokens is the limited strategy used by existing methods [4][5], which our work seeks to improve.
> Our approach more comprehensively identifies sink tokens by leveraging the relationship between extreme activation outliers and attention sinks.
>
> **Response to Questions to Authors:**
> 1. To further validate the improvements brought by KVSink on more challenging tasks, we conduct additional experiments on LongBench V2 [6] using KIVI [7], a widely adopted 2-bit KV cache quantization method.
> LongBench V2 is a benchmark designed to assess the ability of LLMs to handle deep understanding and reasoning across real-world multitask scenarios, including tasks such as document QA, long in-context learning, among others.
> As shown in the table, KVSink effectively improves the performance.
> | Model | Method | Overall | +CoT | Easy | +CoT | Hard | +CoT | Short | +CoT | Medium | +CoT | Long | +CoT |
> |-|-|-|-|-|-|-|-|-|-|-|-|-|-|
> | **Llama3-8B** | KIVI-2bit-gs128 | 15.9 | 8.5 | 18.8 | 9.9  | 14.1 | 7.7 | 13.9 | 7.8 | 20.5 | 10.2 | 10.2 | 6.5 |
> | | + KVSink | 16.3 | 11.3 | 17.2 | 10.4 | 15.8 | 11.9 | 13.9 | 12.2 | 20.5 | 11.6 | 12.0 | 9.3 |
> | **Llama3-8B-instruct** | KIVI-2bit-gs128 | 25.0 | 19.7 | 24.5 | 17.7 | 25.4 | 20.9 | 28.3 | 20.0 | 24.2 | 17.2 | 21.3 | 24.1 |
> | | + KVSink | 26.2 | 19.3 | 27.6 | 21.4 | 25.4 | 18.0 | 29.4 | 18.3 | 23.7 | 19.5 | 25.9 | 20.4 |
> 2. The experiments in the paper already cover 2-bit and 4-bit quantization.
> For additional evidence of INT2 KV cache quantization across more tasks, please refer to the table in the preceding section. Thank you very much!
> 3. As discussed in Section 4.3 of the paper, we find that dynamically identifying attention sinks is impractical.
> Commonly used efficient attention implementations such as [8], which are based on optimized CUDA kernels, do not expose the intermediate attention weights required to identify attention sinks.
> Our method effectively addresses this issue by leveraging the relationship between extreme activation outliers and attention sinks.
> 4. Thank you for raising this important point regarding the limitations of our work.
> Existing theoretical studies on attention sinks [1][2][3], including ours, primarily focus on dense architectures.
> Research on attention sinks within Mixture of Experts architectures remains limited.
> Validating attention sinks and the effectiveness of KVSink across a broader range of attention-based models is an ongoing direction of our research.
>
> If our response meets your expectations, we would kindly appreciate it if the rating could be reconsidered.
> We remain happy to address any further concerns. Thank you very much!
>
> **Reference:**
> [1] Efficient Streaming Language Models with Attention Sinks
>
> [2] Massive Activations in Large Language Models
>
> [3] When Attention Sink Emerges in Language Models: An Empirical View
>
> [4] KVQuant: Towards 10 Million Context Length LLM Inference with KV Cache Quantization
>
> [5] SKVQ: Sliding-window Key and Value Cache Quantization for Large Language Models
>
> [6] LongBench v2: Towards Deeper Understanding and Reasoning on Realistic Long-context Multitasks
>
> [7] KIVI: A Tuning-Free Asymmetric 2bit Quantization for KV Cache
>
> [8] FlashAttention-3: Fast and Accurate Attention with Asynchrony and Low-precision

---

### Official Review · Reviewer_Uszr · 2025-05-14

**Rating:** 7
**Confidence:** 3
**Ethics Flag:** 1

**Summary:**

This paper introduces KVSink, a novel method for predicting sink tokens to enable their more thorough preservation during quantization. The work highlights the observation that attention sinks are not limited to initial token positions and that their effective preservation is crucial for maintaining model performance post-quantization. The paper presents the detailed analysis of extreme activation outliers and demonstrates the effectiveness of KVSink in preserving performance on downstream tasks.

**Questions To Authors:**

1. Does KVSink generalize well to a wider variety of LLM models, such as the Qwen series models.
2. How robust is the KVSink method to varying sequence lengths? Are any limitations observed?
3. How robust is the KVSink method to different types of LLM architectures, such as encoder-decoder architectures.
4. Are the evaluation datasets, WikiText-2 and C4 enough? The paper KVQuant includes long context length evaluation.

**Reasons To Accept:**

1. Efficient LLM inference is a critical area of research. This paper focuses on attention sinks and tackles the important challenge of performance degradation in LLMs due to KV cache quantization.
2. This paper introduces a practical, plug-and-play method (KVSink) that can effectively predict and preserve these crucial tokens with low overhead.
3. The authors provide extensive experimental results across various models and datasets to demonstrate the effectiveness of the proposed approach.

**Reasons To Reject:**

See Questions to Authors below.

---

> ### Author Response · Authors · 2025-05-31
>
> We sincerely appreciate the time and effort that Reviewer Uszr devoted to evaluating our work.
> The thoughtful and constructive feedback will be very helpful to our future research.
> Below, we will make every effort to address the concerns raised.
>
> **Response to Questions to Authors:**
> 1. To address Questions 1, 2, and 4, we further evaluate the effectiveness of KVSink across a wider range of models, context lengths, and datasets.
> Specifically, we conduct additional experiments on LongBench V2 [1] using KIVI [2], a widely adopted 2-bit KV cache quantization method.
> LongBench V2 is a benchmark designed to evaluate the capability of LLMs in handling long-context tasks that require deep understanding and reasoning across real-world, multitask scenarios, including document QA, long in-context learning, and others.
> The corresponding emergence layers and outlier channels used in the experiments are detailed in the second table.
> | Model | Method | Overall | +CoT | Easy | +CoT | Hard | +CoT | Short | +CoT | Medium | +CoT | Long | +CoT |
> |-|-|-|-|-|-|-|-|-|-|-|-|-|-|
> | **Llama3-8B** | KIVI-2bit-gs128 | 15.9 | 8.5 | 18.8 | 9.9  | 14.1 | 7.7 | 13.9 | 7.8 | 20.5 | 10.2 | 10.2 | 6.5 |
> | | + KVSink | 16.3 | 11.3 | 17.2 | 10.4 | 15.8 | 11.9 | 13.9 | 12.2 | 20.5 | 11.6 | 12.0 | 9.3 |
> | **Llama3-8B-instruct** | KIVI-2bit-gs128 | 25.0 | 19.7 | 24.5 | 17.7 | 25.4 | 20.9 | 28.3 | 20.0 | 24.2 | 17.2 | 21.3 | 24.1 |
> | | + KVSink | 26.2 | 19.3 | 27.6 | 21.4 | 25.4 | 18.0 | 29.4 | 18.3 | 23.7 | 19.5 | 25.9 | 20.4 |
> | **Qwen2.5-3B** | KIVI-2bit-gs128 | 20.9 | 24.1 | 18.8 | 21.4 | 22.2 | 25.7 | 18.3 | 28.3 | 17.2 | 22.8 | 32.4 | 19.4 |
> | | + KVSink | 22.9 | 26.4 | 18.2 | 28.1 | 25.7 | 25.4 | 21.7 | 27.2 | 21.4 | 26.5 | 27.8 | 25.0 |
> | **Qwen2.5-7B** | KIVI-2bit-gs128 | 19.9 | 27.4 | 22.4 | 30.7 | 18.3 | 25.4 | 25.6 | 28.3 | 14.0 | 24.7 | 22.2 | 31.5 |
> | | + KVSink | 22.3 | 28.2 | 27.1 | 29.7 | 19.3 | 27.3 | 23.3 | 30.6 | 20.5 | 27.0 | 24.1 | 26.9 |
> 2. Thank you for raising this insightful and forward-looking question regarding the applicability of our method to different LLM architectures.
> To the best of our knowledge, theoretical investigations of attention sinks in LLMs have thus far been limited to decoder-only dense architectures.
> Attention sinks in other architectures, such as encoder-decoder LLMs or Mixture-of-Experts (MoE) LLMs, remain largely unexplored.
> Extending this line of research and evaluating our method across diverse architectures is a key direction of our future work.
> We have already conducted preliminary investigations on MoE LLMs, including DeepSeek-V3/R1 and MoE models from the Qwen3 series, where the presence of attention sinks has been verified.
> The corresponding emergence layers and outlier channels identified in these models are presented in the table.
> We sincerely appreciate your insightful question and will continue to refine and extend our approach in future research.
> | Model  | Emergence Layer | Outlier Channel |
> |-|-|-|
> |DeepSeek-V3|15|6387|
> |DeepSeek-R1|15|6387|
> |Qwen3-30B-A3B|4|940|
> |Qwen3-235B-A22B|4|1806|
> |Llama3-8B|1|788|
> |Llama3-8B-instruct|1|788|
> |Qwen2.5-3B-instruct|3|318|
> |Qwen2.5-7B-instruct|4|458|
>
> If our response meets your expectations, we would kindly appreciate it if the rating could be reconsidered.
> We remain happy to address any further concerns. Thank you very much!
>
> **Reference:**
> [1] LongBench v2: Towards Deeper Understanding and Reasoning on Realistic Long-context Multitasks
>
> [2] KIVI: A Tuning-Free Asymmetric 2bit Quantization for KV Cache

---

### Official Review · Reviewer_G7Pw · 2025-05-14

**Rating:** 6
**Confidence:** 4
**Ethics Flag:** 1

**Summary:**

The paper analyzes why preserving only the first few tokens during KV-cache quantization is insufficient by tracing how attention sink tokens propagate extreme activation outliers through decoder layers. it shows that sink tokens can arise anywhere in the context and that their suppressed QKV norms plus high query-key similarity create implicit attention biases that are highly sensitive to quantization. The authors introduce KVSink, a lightweight procedure that detects stable activation outliers in a single early layer and uses their positions to mark sink tokens whose KV pairs must be kept at higher precision. Experiments on seven open-source Llama and Mistral models demonstrate that protecting just five KVSink tokens consistently lowers perplexity relative to the traditional Preserve-First-N strategy and further improves the state-of-the-art KVQuant method.

**Questions To Authors:**

How does KVSink behave during streaming or speculative decoding, where sink tokens might appear after the prefill phase and the emergence layer has already passed; can the method be run without noticeable latency?

**Reasons To Accept:**

The work provides explanation of the causal link between activation outliers, attention sinks, and quantization error, provides subsequent design choices in clear empirical evidence; KVSink is model-agnostic, plug-and-play, and requires only a top-k magnitude scan in one pre-identified channel, so its computational and memory costs are almost zero at long context lengths.

**Reasons To Reject:**

All reported benefits are measured in perplexity; The paper does not provide end-to-end decoding throughput or GPU memory results for real-time inference, so practical impact is unquantified. KVSink relies on a manually tabulated emergence layer and outlier channel for each model, which may need re-profiling after fine-tuning or architecture changes.

---

> ### Author Response · Authors · 2025-06-01
> **Official Response by the Authors (Part 1)**
>
> We sincerely appreciate the time and effort that Reviewer G7Pw devoted to evaluating our work.
> The thoughtful and constructive feedback will be very helpful to our future research.
> Below, we will make every effort to address the concerns raised.
>
> **Minor Clarification Regarding the Summary Content:**
> 1. Thank you for the excellent summary of our paper.
> We would like to provide a brief clarification regarding the statement that “sink tokens can arise anywhere in the context” to avoid potential misunderstandings.
> Based on both our observations and existing literature [1, 2], these sinks typically appear near the beginning of the sequence, aligning with first token, semantically weak tokens or delimiter tokens.
> We use diverse samples from the C4 and WikiText-2 datasets to illustrate this phenomenon, as summarized in the table.
> | Model | Input | Sink Token | Token Index | Total Input Tokens | Channel Index |
> |-|-|-|-|-|-|
> |Llama2-7B|sample 1|"."|13|2048|2533|
> ||sample 2|"."|6|2048|2533|
> ||sample 3|"."|4|2048|2533|
> |Llama2-7B-Chat|sample 1|"."|13|2048|2533|
> ||sample 2|"."|6|2048|2533|
> ||sample 3|"."|4|2048|2533|
> |Mistral-7B|sample 1|"in"|3|2048|2070|
> ||sample 2|","|2|2048|2070|
> ||sample 3|"\n"|5|2048|2070|
>
> **Reference:**
> [1] Massive Activations in Large Language Models
>
> [2] Unveiling and Harnessing Hidden Attention Sinks

---

> > ### Author Response · Authors · 2025-06-01
> > **Official Response by the Authors (Part 2)**
> >
> > **Response to Reasons To Reject:**
> > 1. we further evaluate KVSink across a wider range of models, context lengths, and datasets.
> > Specifically, we conduct additional experiments on LongBench V2 [4] using KIVI [5].
> > LongBench V2 is a benchmark designed to evaluate the capability of LLMs in handling long-context tasks that require deep understanding and reasoning across real-world, multitask scenarios.
> > | Model | Method | Overall | +CoT | Easy | +CoT | Hard | +CoT | Short | +CoT | Medium | +CoT | Long | +CoT |
> > |-|-|-|-|-|-|-|-|-|-|-|-|-|-|
> > | **Llama3-8B** | KIVI-2bit-gs128 | 15.9 | 8.5 | 18.8 | 9.9  | 14.1 | 7.7 | 13.9 | 7.8 | 20.5 | 10.2 | 10.2 | 6.5 |
> > | | + KVSink | 16.3 | 11.3 | 17.2 | 10.4 | 15.8 | 11.9 | 13.9 | 12.2 | 20.5 | 11.6 | 12.0 | 9.3 |
> > | **Llama3-8B-instruct** | KIVI-2bit-gs128 | 25.0 | 19.7 | 24.5 | 17.7 | 25.4 | 20.9 | 28.3 | 20.0 | 24.2 | 17.2 | 21.3 | 24.1 |
> > | | + KVSink | 26.2 | 19.3 | 27.6 | 21.4 | 25.4 | 18.0 | 29.4 | 18.3 | 23.7 | 19.5 | 25.9 | 20.4 |
> >
> > 2. We conduct additional efficiency tests using Llama3-8B on a single 80GB A100 GPU.
> > Specifically, we evaluate performance under input lengths of 2K, 4K, and 8K tokens with 100 generated output tokens, and report the prefill time, average decoding time, and peak GPU memory consumption.
> > As shown in the table, the additional overhead introduced by KVSink remains negligible, since it performs lightweight identification in a single emergence layer during the prefill stage and retains only a small number of sink tokens.
> > |Model| Method |Input Length|Decode Length|Prefill time (ms)|Average Decoding time (ms)|Peak GPU Memory (GB)|
> > |-|-|-|-|-|-|-|
> > |Llama-3-8B|KIVI|2K|100|1217.17|56.90|16.492|
> > ||+KVSink|2K|100|1257.87|56.59|16.494|
> > ||KIVI|4K|100|1343.55|57.16|18.009|
> > ||+KVSink|4K|100|1391.75|56.82|18.012|
> > ||KIVI|8092|100|1548.56|58.39|20.978|
> > ||+KVSink|8092|100|1611.02|58.02|20.980|
> >
> > 3. Thank you for raising this important question.
> > First, prior studies on attention sinks [3] and extreme activation outliers [1] have shown that fine-tuning does not alter the distribution of attention sinks or outlier channels, as confirmed by our experimental results in the table.
> > Second, since emergence layers and outlier channels are input-independent, identifying them for a new model architecture requires only a single forward pass.
> > | Model  | Emergence Layer | Outlier Channel |
> > |-|-|-|
> > |DeepSeek-V3|15|6387|
> > |DeepSeek-R1|15|6387|
> > |Llama3-8B|1|788|
> > |Llama3-8B-instruct|1|788|
> > |Qwen2.5-7B|4|458|
> > |Qwen2.5-7B-instruct|4|458|
> >
> > **Response to Questions To Authors:**
> > 1. We sincerely appreciate this insightful and important question. To provide a clear response, we divide our explanation into two aspects:
> > (1) The emergence of sink tokens after the prefill stage; and
> > (2) The behavior of KVSink during streaming or speculative decoding.
> >
> > For (1), we acknowledge that while our work advances the understanding of attention sinks and enables more comprehensive identification of sink tokens compared to traditional preserve-first-n strategies, it still shares the limitation of not capturing sink tokens that emerge after the prefill stage.
> > This design decision is informed by empirical findings showing that sink tokens occur near the beginning of the input sequence.
> > In practice, the model’s initial prompt (e.g., system instructions or early conversation history) is typically long enough to trigger the emergence of these extremely sparse sink tokens.
> > To the best of our knowledge, existing studies on attention sinks lack theoretical evidence for this behavior.
> > We hypothesize that, as one of the roles of attention sinks is to introduce an attention bias [2,4], they are most effective when positioned near the beginning of the sequence, where they can influence the majority of subsequent tokens.
> > We leave a rigorous theoretical investigation of this phenomenon to future work.
> >
> > For (2), our analysis of attention sinks focuses on their general behavioral patterns and is not tied to any specific application scenario.
> > As such, the identification method remains broadly applicable across various LLM use cases.
> > For streaming or speculative decoding, as discussed in (1), while we cannot rule out the possibility that additional attention sinks may emerge after the prefill stage, our method still offers a more effective identification of attention sinks compared to the naive preserve-first-n strategy.
> >
> > 2. Please refer to Point 2 of our Response to Reasons for Rejection.
> >
> > If our response meets your expectations, we would kindly appreciate it if the rating could be reconsidered.
> > We remain happy to address any further concerns. Thank you very much!
> >
> > **Reference:**
> > [1] Massive Activations in Large Language Models
> >
> > [2] Unveiling and Harnessing Hidden Attention Sinks
> >
> > [3] When Attention Sink Emerges in Language Models: An Empirical View
> >
> > [4] Towards Deeper Understanding and Reasoning on Realistic Long-context Multitasks
> >
> > [5] A Tuning-Free Asymmetric 2bit Quantization for KV Cache

---

> > > ### Comment · Reviewer_G7Pw · 2025-06-05
> > >
> > > Thanks to the authors on the response on the KVSink behavior and additional efficiency tests. i increase score to 6

---

### Official Review · Reviewer_vwcZ · 2025-05-18

**Rating:** 6
**Confidence:** 3
**Ethics Flag:** 1

**Summary:**

This paper analyze the phenomenon of attention sink and explores its interaction with KV cache quantization. Building on the understanding of stable outliers, the authors propose an efficient method to predict sink tokens with minimal computation. Experimental results demonstrate the effectiveness of the proposed approach.

**Reasons To Accept:**

The paper study the attention sink phenomenon. This is an underexplored but significant behavior in LLMs. Studying its effect on KV cache quantization is a novel and meaningful direction, especially given the inherent trade-off between range and precision during quantization. The authors highlight how per-channel quantization can lead to large errors due to attention sink, and propose an algorithm that efficiently predicts these sink tokens by leveraging insights from stable outliers. The paper is insightful and well-written. Figures are illustrative.

**Reasons To Reject:**

The evaluation is somewhat limited. The paper only compare against the Preserve-First-N baseline. Given the context, it would be more informative to include comparisons with additional baselines that also preserve a fixed number of tokens, which are common in KV cache pruning.

---

> ### Author Response · Authors · 2025-05-31
>
> We sincerely appreciate the time and effort that Reviewer vwcZ devoted to evaluating our work.
> The thoughtful and constructive feedback will be very helpful to our future research.
> Below, we will make every effort to address the concerns raised.
>
> **Response to Reasons To Reject:**
> 1. Thank you for raising this thoughtful and insightful question.
> Our work primarily focuses on analyzing the relationship between KV cache quantization and attention sinks, including a detailed investigation of how attention sinks impact various quantization schemes.
> This is why our experiments primarily evaluate quantization-based methods.
> As you have insightfully noted, our method is also applicable to KV cache pruning strategies that preserve a fixed number of initial tokens.
> We further evaluate our method on StreamingLLM [1] using LongBench V2 [2], a benchmark designed to assess the long-context understanding and reasoning capabilities of LLMs across diverse real-world tasks.
> StreamingLLM safeguards attention sinks by preserving a fixed number of initial tokens; however, this strategy does not account for sink tokens that may emerge in other positions.
> In our experiments, we set the number of preserved initial or identified sink tokens to 4, and retained 1020 local tokens.
> As shown in the table, our method helps address this limitation and leads to improved performance.
> | Model | Method | Overall | +CoT | Easy | +CoT | Hard | +CoT | Short | +CoT | Medium | +CoT | Long | +CoT |
> |-|-|-|-|-|-|-|-|-|-|-|-|-|-|
> | **Llama3-8B** | StreamingLLM | 13.9 | 11.1 | 15.1 | 14.6 | 13.2 | 9.0 | 9.4 | 10.0 | 17.2 | 9.8 | 14.8 | 15.7 |
> | | + KVSink | 14.0 | 12.1 | 11.8 | 15.8 | 15.3 | 9.7 | 8.3 | 9.0 | 18.7 | 11.4 | 13.2 | 18.8 |
>
> We sincerely appreciate this valuable suggestion and will further evaluate our method across a broader range of approaches and tasks in future work.
> If our response meets your expectations, we would kindly appreciate it if the rating could be reconsidered.
> We remain happy to address any further concerns. Thank you very much!
>
> **Reference:**
> [1] Efficient Streaming Language Models with Attention Sinks
>
> [2] LongBench v2: Towards Deeper Understanding and Reasoning on Realistic Long-context Multitasks

---

### Author Response · Authors · 2025-06-09

Dear Reviewers,

As the discussion period comes to a close, we remain available to address any additional concerns you may have. If our previous response aligns with your expectations, we would be grateful if you could kindly reconsider the rating.

We sincerely appreciate the time and effort you have devoted to reviewing our work. Thank you once again.

---

### Decision · Program_Chairs · 2025-07-08

**Decision:**

Accept

**Comment:**

This paper presents KVSink, a novel method for identifying and preserving attention sink tokens to improve the quality of KV cache quantization in LLMs. The work is motivated by the observation that attention sinks — tokens that disproportionately attract attention — are not necessarily among the initial tokens and are particularly vulnerable to precision loss under quantization. KVSink leverages the emergence of stable activation outliers to efficiently identify such tokens early, resulting in minimal overhead and consistent gains in perplexity across multiple models.

Overall evaluation:
- The technical depth and rigor of the analysis are strong. The authors provide a compelling explanation of how attention sinks and outlier activations interact with quantization errors.
- The paper is clearly written, with well-structured explanations and informative figures. Reviewers appreciated the clarity of the causal link between activation outliers, attention behavior, and quantization error.
- The paper explores a novel and underexamined issue in LLM optimization: the role of attention sinks in quantization artifacts.
- Efficient inference in LLMs is a highly relevant research area, and KVSink provides a practical contribution with potential downstream impact.

The paper can be strengthened in the following aspects:
- Evaluation is limited to perplexity; Adding the results on LongBench during the discussion phase to the main paper is helpful. Also, provide end-to-end decoding throughput or GPU memory results for real-time inference.
- Justify the generalization to other model families (e.g., Qwen, encoder-decoder)
- Justify some heuristics (e.g., pre-defined emergence layers and outlier channels).

This is a high-quality and insightful paper that addresses an important efficiency problem in LLM deployment. Despite some missing evaluations and questions about generalizability, its conceptual contribution and empirical effectiveness reach the acceptance bar.